# Widespread human exposure to ledanteviruses in Uganda: A population study

**James G. Shepherd**[1]*, **Shirin Ashraf**[1], **Jesus F. Salazar-Gonzalez**[2], **Maria G. Salazar**[3], **Robert G. Downing**[4], **Henry Bukenya**[4], **Hanna Jerome**[1], **Joseph T. Mpanga**[4], **Chris Davis**[1], **Lily Tong**[1], **Vattipally B. Sreenu**[1], **Linda A. Atiku**[4], **Nicola Logan**[1], **Ezekiel Kajik**[4], **Yafesi Mukobi**[4], **Cyrus Mungujakisa**[4], **Michael V. Olowo**[4], **Emmanuel Tibo**[4], **Fred Wunna**[4], **Hollie Jackson Ireland**[1], **Andrew E. Blunsum**[1], **Iyanuoluwani Owolabi**[1], **Ana da Silva Filipe**[1], **Josephine Bwogi**[4], **Brian J. Willett**[1], **Julius J. Lutwama**[4], **Daniel G. Streicker**[1], **Pontiano Kaleebu**[3,4], **Emma C. Thomson**[1,5]*

1 Medical Research Council-University of Glasgow Centre for Virus Research, Glasgow, United Kingdom, 2 National Institute of Allergy and Infectious Diseases, National Institutes of Health, Bethesda, Maryland, United States of America, 3 Medical Research Council/Uganda Virus Research Institute and London School of Hygiene & Tropical Medicine Uganda Research Unit, Entebbe, Uganda, 4 Uganda Virus Research Institute, Entebbe, Uganda, 5 London School of Hygiene and Tropical Medicine, London, United Kingdom

* james.shepherd.2@glasgow.ac.uk (JGS); emma.thomson@glasgow.ac.uk (ECT)

**Data Availability Statement:** Consensus viral genome and mitochondrial cytochrome B sequences have been submitted to GenBank under accession numbers OQ077988, OR497399 and

## Abstract

Le Dantec virus (LDV), assigned to the species *Ledantevirus ledantec*, genus *Ledantevirus*, family *Rhabdoviridae* has been associated with human disease but has gone undetected since the 1970s. We describe the detection of LDV in a human case of undifferentiated fever in Uganda by metagenomic sequencing and demonstrate a serological response using ELISA and pseudotype neutralisation. By screening 997 individuals sampled in 2016, we show frequent exposure to ledanteviruses with 76% of individuals seropositive in Western Uganda, but lower seroprevalence in other areas. Serological cross-reactivity as measured by pseudotype-based neutralisation was confined to ledanteviruses, indicating population seropositivity may represent either exposure to LDV or related ledanteviruses. We also describe the discovery of a closely related ledantevirus in blood from the synanthropic rodent *Mastomys erythroleucus*. Ledantevirus infection is common in Uganda but is geographically heterogenous. Further surveys of patients presenting with acute fever are required to determine the contribution of these emerging viruses to febrile illness in Uganda.

## Author summary

Understanding the viruses capable of human infection is important for outbreak prevention and early intervention in viral epidemics. Le Dantec virus (LDV) is a member of the viral genus *Ledantevirus (Rhabodoviridae)* and has previously been isolated only once previously in a child with febrile illness in Senegal in 1965. We detected the genome of LDV in blood sampled from a patient presenting with febrile illness in Western Uganda in 2012. To estimate the extent to which LDV may be causing human infection in the region, we tested stored blood samples collected in 2016 for evidence of antibodies to LDV,

OR607590. Raw sequencing read files derived from rodent blood have been submitted to NCBI sequence read archive (BioProject ID PRJNA1013481, Biosample accession SAMN37299165). Experimental data relating to LEDV-ELISA and pseudotype-based neutralisation assays have been provided as supporting information files.

**Funding:** This work was funded by awards from the Wellcome Trust (102789/Z/13/Z to ET and 217221/Z/19/Z to DS, https://wellcome.org/) and the Medical Research Council (MC_UU_12014/8 to ET and MC_ST_U17020 to JS, https://www.ukri.org/councils/mrc/). The funders had no role in study design, data collection and analysis, decision to publish, or preparation of the manuscript.

**Competing interests:** The authors have declared that no competing interests exist.

revealing that up to 76% of Ugandans in some areas had previously been exposed to either to LDV or a closely related virus. Seroprevalence was highest in Western Uganda, an area of high biodiversity where several other ledanteviruses have been isolated in association with ectoparasites of bats. To investigate potential ecological reservoirs of zoonotic viruses we tested blood from wild rodents inhabiting areas close to human settlements. We detected a new ledantevirus, closely related to LDV, in blood from a wild rodent *Mastomys erythroleucus*. Our work shows that ledanteviruses are a common cause of human infection in Uganda.

## Introduction

The epidemiology of non-malarial acute febrile illness in sub-Saharan Africa is poorly characterised [1]. Despite the presence of a variety of endemic human pathogens and zoonoses, the diagnostic infrastructure in much of Africa is limited in comparison to the extensive culture-based and molecular assays available to patients in high income countries [2]. Paradoxically, the existing diversity of human pathogens in sub-Saharan Africa is likely to expand further as human populations encroach into novel ecosystems [3].

Viruses contribute heavily to common fever-related presentations in Africa [4], but many research studies perform limited viral diagnostics or omit them entirely [5]. As such there is an incomplete understanding of the viruses causing febrile illness in Africa, limiting the potential for early epidemic and pandemic response. To address this, researchers have employed unbiased metagenomic Next generation sequencing (mNGS) for pathogen diagnosis in cohorts of patients with acute febrile illness. This approach involves amplification of all nucleic acids within a sample in a "pathogen agnostic" manner and has led to the discovery several novel viruses associated with human disease [6–8].

Le Dantec virus (LDV), assigned to the species *Ledantevirus ledantec*, genus *Ledantevirus*, family *Rhabdoviridae*, was first isolated in association with human febrile illness in Senegal in 1965 [9]. Here we describe detection of LDV in the blood of a Ugandan patient with undifferentiated acute febrile illness in 2012 by mNGS. We demonstrate a serological response against the viral glycoprotein in patient serum by ELISA, immunocytochemistry and pseudotype-based neutralisation. By screening stored serum samples from 2016 by ELISA, we show evidence of significant but geographically heterogenous exposure to LDV across Uganda, with seropositivity highest in a region notable for previous detection of ledanteviruses in ectoparasites of bats. Testing ELISA-reactive sera against a panel of pseudotype viruses expressing the glycoproteins of related ledanteviruses revealed that in addition to LDV, other closely related viruses are likely to be causing human infection via zoonotic transmission. In support of this possibility, we report the detection of a putative novel ledantevirus, closely related to LDV, in a rodent host derived from peri-domestic sampling in Northern Uganda.

## Results

### Patient characteristics

The findings of the Acute Febrile Illness (AFI) study including the detection of LDV have been reported previously [10,11]. Here we focus on the clinical presentation of the LDV patient. The patient was a 9 year-old male residing in Kasese district, Western Uganda, who was recruited to the AFI study, an observational study investigating the epidemiology of acute febrile illness in Uganda. He presented to his local health centre in May 2012 with a 4-day

history of fever, chills, headache, arthralgia, abdominal pain, and vomiting. He had a fever of 38.5°C, but no other abnormal findings on physical examination. A presumptive diagnosis of typhoid was made by the treating physician based on clinical features and he was managed as an outpatient with oral ciprofloxacin. He had fully recovered at a follow-up appointment four weeks later and at further follow up in 2018 he remained well. After enrolment in the AFI study, blood samples were tested against multiple pathogens including leptospirosis, brucellosis, malaria, dengue, chikungunya, yellow fever, o'nyong nyong, typhoid and rickettsiosis, all of which were negative (S1 Table).

## Detection of LDV by metagenomic sequencing of acute patient samples

As initial diagnostic investigations were negative, a serum sample from the acute phase of the patient's illness was subjected to mNGS. Extracted RNA from serum was sequenced on an Illumina MiSeq instrument, resulting in 1677574 paired-end sequencing reads. *De novo* assembly yielded a 11,444 nucleotide contig that aligned to the genome of LDV on blastn query of the NCBI nucleotide database (isolate DakHD763, KM205006, 11450 base pairs). This contig was used as a reference sequence for alignment of raw sequencing reads, resulting in alignment of 8958 reads with 99.7% genome coverage at a minimum depth of 10 reads. Pairwise alignment of the final genome showed greater than 90% nucleotide similarity to the original LDV isolate DakHD763 in all genomic regions (S2 Table). Phylogenetic analysis confirmed the placement of the Ugandan LDV isolate within the genus *Ledantevirus* (Fig 1).

## Confirmation of LDV infection by PCR and screening of acute fever samples

To confirm mNGS discovery of LDV, a freshly prepared RNA extract from patient serum was assayed by RT-PCR with primers designed against conserved nucleotide positions in the original DakHD763 isolate and the sequence derived from Ugandan patient. The PCR assay yielded a 5.4 Kb fragment. A separate RT-PCR assay targeting a 2.4 Kb fragment in the glycoprotein gene was also positive. Sanger sequencing of the 5.4 Kb amplicon revealed all but one of the 5,406 nucleotides to be identical to the metagenomic sequence, validating the metagenomic sequence and confirming the presence of LDV RNA in the serum of the Ugandan patient.

The RNA was also tested with a separate LDV-specific real-time RT-PCR assay. Using serial 10-fold dilutions of the DakHD763 isolate RNA as positive control we detected the 145-bp LDV sequence with a linear dynamic range of over four orders of magnitude. A fresh RNA extract from the source patient also showed a detectable signal; three µl of plasma-equivalent RNA yielded a Ct value of 30, consistent with a substantial viremia. Plasma samples (n = 62) from a measles/rubella cohort and plasma pools (10 per pool) of unexplained acute febrile infections from Sudan tested all negative for LDV RNA in the real-time RT-PCR assay.

## Demonstration of a serological response to infection with LDV in the index patient

Using an in-house LDV glycoprotein (LDV-G) ELISA we demonstrated reactivity to LDV-G by both purified IgG and serum from blood samples collected from the patient at a follow-up visit in 2018 (Fig 2). Reactivity was also detected in a close relative who had experienced a contemporaneous illness but had not presented to hospital. A further 10 unrelated serum samples from Uganda and 12 from healthy UK controls were tested, with evidence of an antibody response to LDV-G detectable in both the index case and their close contact as well as two

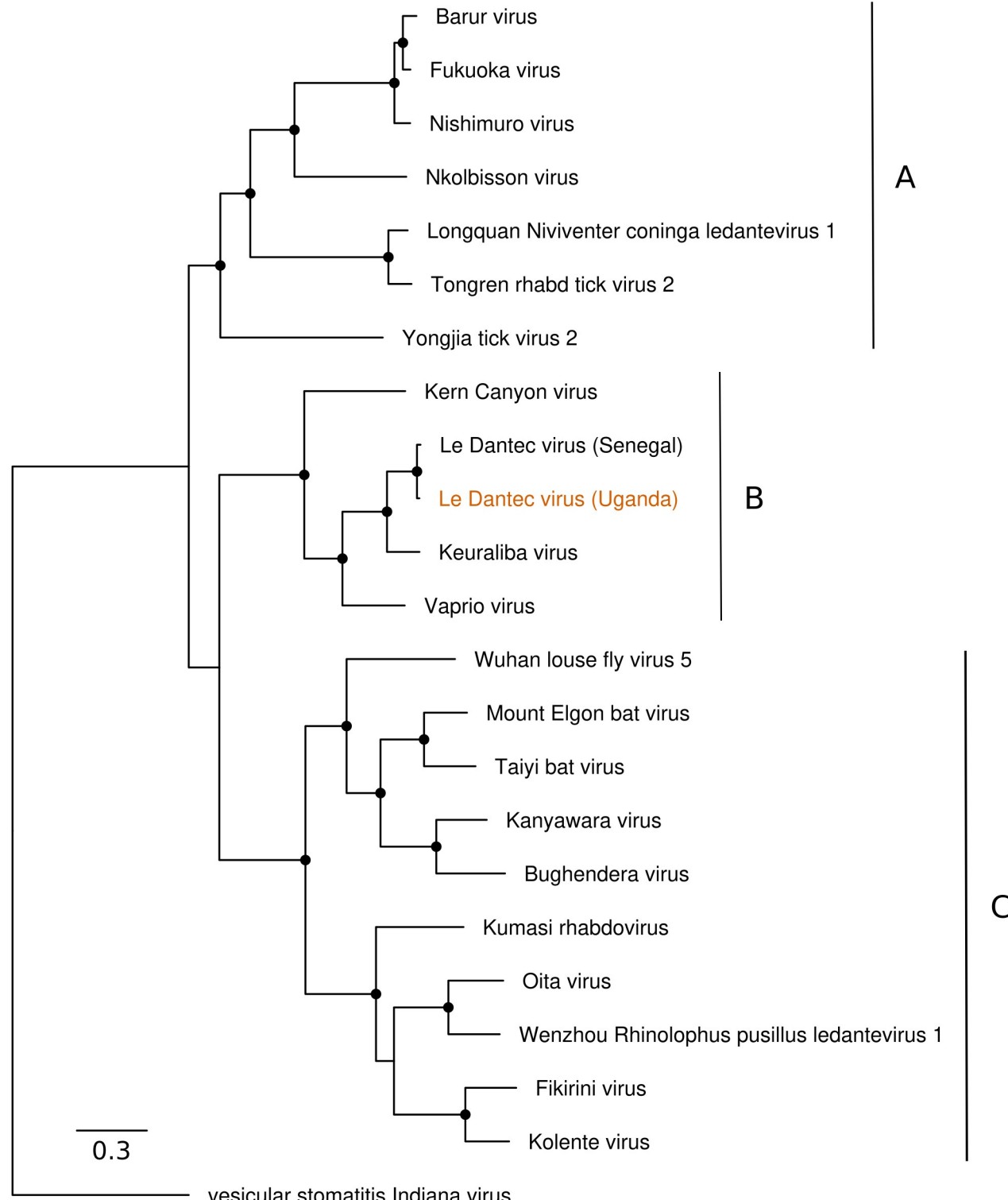

**Fig 1. Phylogenetic relationships of Le Dantec Virus identified in Uganda.** Maximum likelihood phylogeny of the genus *Ledantevirus* based on full-length L protein amino acid sequences. The phylogroups A, B and C indicated to the right. Vesicular stomatitis Indiana virus is included as an outgroup. The sequence derived from the febrile patient described in this study is indicated in orange text. The scale bar represents substitutions per site. Nodes with bootstrap support >70 are indicated with black circles.

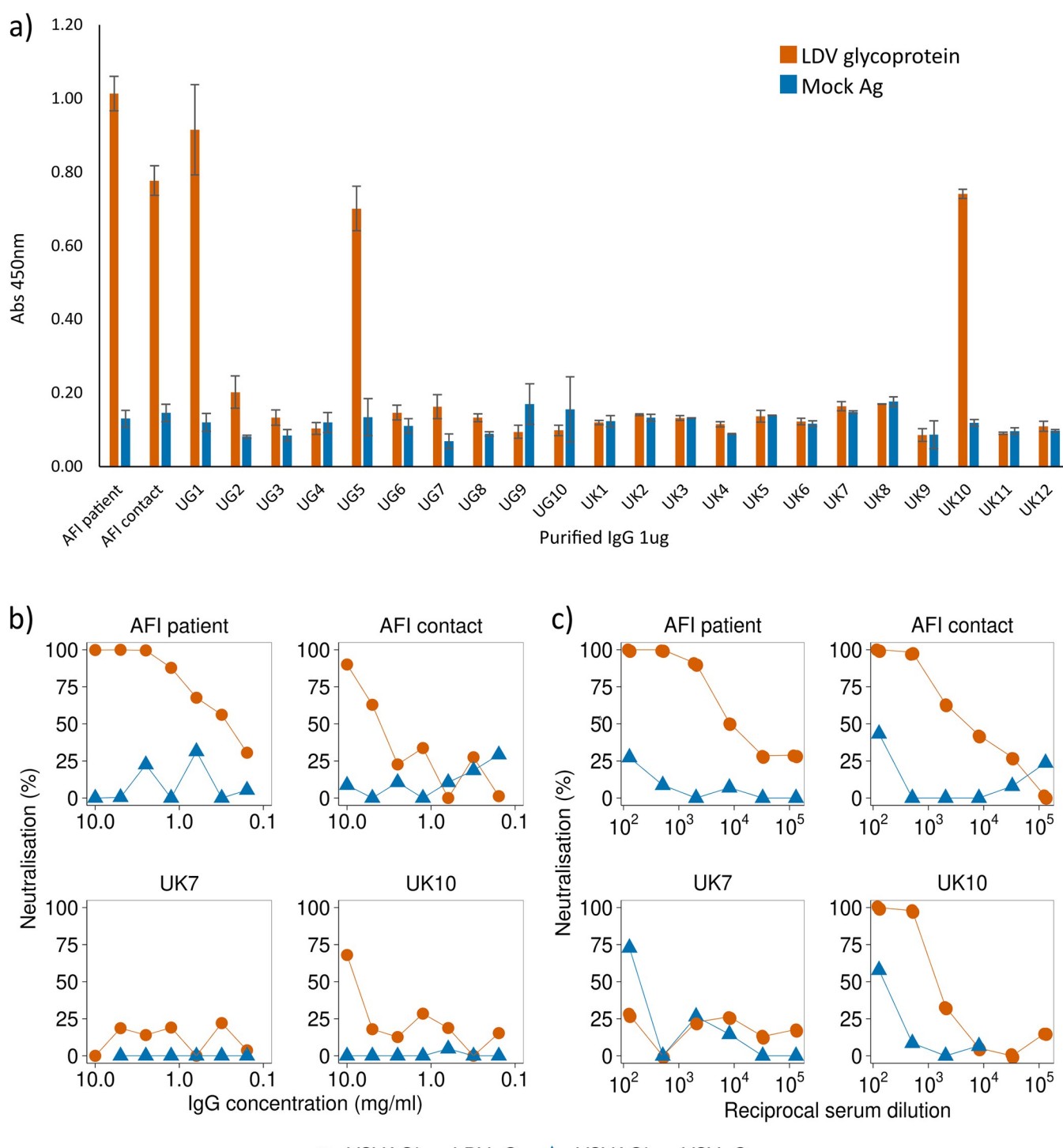

**Fig 2. Serological detection of LDV in patient serum by ELISA and pseudotype neutralisation.** a) Raw OD450 values for purified IgG from the index patient, a close contact and Ugandan and UK controls tested by LDV-G ELISA. Bars represent the mean OD450 based on six technical replicates. Error bars represent the standard error of the mean. b) Neutralisation of VSVΔ*Gluc*-LDV-G and VSVΔ*Gluc*-VSV-G by purified IgG from human serum. Serial IgG dilutions were mixed with pseudotype preparations and added to HEK293T cells. Luciferase activity was measured at 48 hours post infection. Datapoints represent the mean neutralisation derived from three technical replicates. c) Neutralisation of VSVΔ*Gluc*-LDV-G and VSVΔ*Gluc*-VSV-G by human serum. Serial serum dilutions were mixed with pseudotype preparations and added to HEK293T cells. Luciferase activity was measured at 48 hours post infection. A representative experiment is shown. Datapoints represent the percentage neutralisation derived from the mean of three technical replicates. Percentage neutralisation represents luciferase readings relative to no-serum control wells.

individuals from Uganda and one sample from the UK (UK10). On further investigation it emerged that UK10 had previously resided in Africa.

A vesicular stomatitis virus (VSV) pseudotype virus with the glycoprotein gene replaced with the gene for firefly luciferase (VSVΔG*luc*) was created to express LDV-G as an outer membrane protein (VSVΔG*luc*-LDV-G). Both serum and purified IgG from the febrile patient specifically inhibited VSVΔG*luc*-LDV-G, without a similar effect on pseudotype viruses displaying the VSV glycoprotein (VSVΔG*luc*-VSV-G, Fig 2B). Consistent with the LDV-G ELISA, serum and purified IgG from the close family contact and UK10 inhibited VSVΔG*luc*-LDV-G, although to a lesser degree than the patient.

To visualise the interactions between LDV-G and patient IgG, LDV-G engineered with a 6-histidine C-terminal tag (LDV-Ghis) was expressed in BHK-21 cells. Cells were incubated with a mixture of a rabbit anti-histidine primary antibody and patient sera. Co-localisation of the rabbit anti-histidine and IgG from patient samples was demonstrated for the patient and other samples found to be positive by ELISA, further demonstrating a serological response directed against LDV-G (Fig 3).

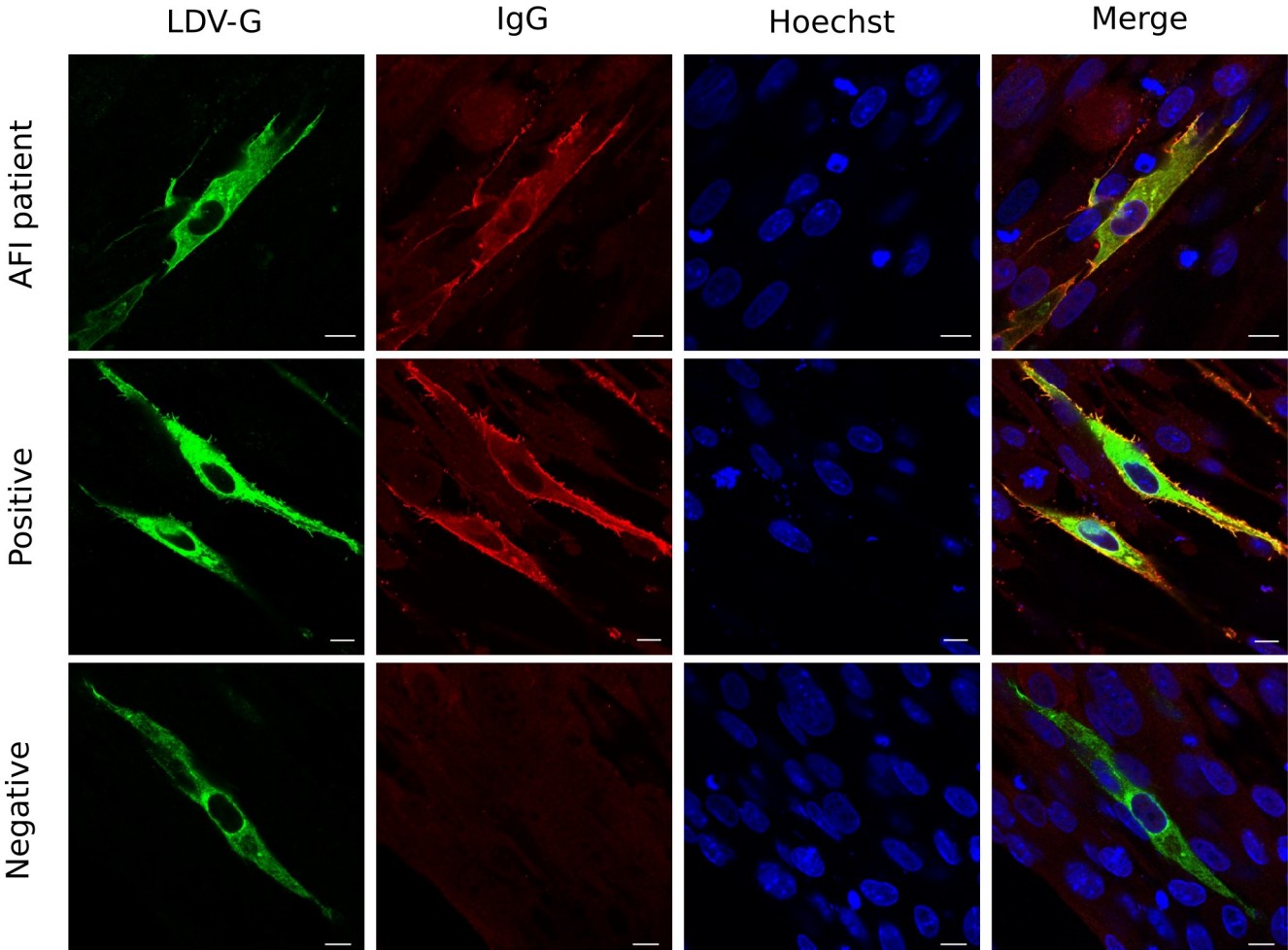

**Fig 3. Co-localisation of LDV-G and human IgG by immunocytochemistry.** BHK-21 cells were transfected with the mammalian expression plasmid VR1012-LDV-Ghis then fixed and permeabilised. Primary antibodies were a rabbit anti-his and human serum samples diluted 1:200. Secondary antibodies were an Alexa Fluror 488 conjugated anti-rabbit antibody (green) and anti-human 594 (red). Each sample was tested in duplicate. A representative experiment including the Ugandan patient serum and positive and negative samples as tested by LDV-G ELISA is shown. Three biological replicates were performed for each sample, with the exception of sera from the index patient where a single replicate was performed owing to limited sample availability.

## High population exposure to LDV in Uganda

To investigate exposure to LDV in the Ugandan population, we screened stored serum samples collected in 2016 as part of the Ugandan national HIV antenatal seroprevalence study (ANC-2016) by LDV-G ELISA. The sample comprised a random selection of sera from 997 women of childbearing age (median age 25, IQR 21–29, range 15–48) from 24 locations across Uganda (Fig 4).

There was heterogeneity in reactivity of serum samples by LDV-G ELISA between the four administrative regions of Uganda (Fig 4). Notably, samples from Northern sites demonstrated low reactivity, in particular the sites in the West-Nile sub-region; Arua, Moyo and Nebbi. As these sites clustered spatially and shared a significantly lower reactivity when compared with other regions, they were considered to represent a low-prevalence population and were used to estimate a positive cut-off value for the assay (mean A450 ratio + 3 standard deviations). Applying this cut-off to all samples we estimated seroprevalence for the different regions of Uganda, with exposure highest in the Western region (75.9%, 95% CI: 71.7–79.8), followed by the Central region (53.3%, 95% CI: 44.9–61.4) and the Eastern region (20.8%, 95% CI: 16.2–26.4). Exposure was markedly lower in the Northern region (3.1%, 95% CI: 1.4–6.5, Fig 4B). The only patient-level metadata recorded for the ANC-2016 dataset was age, which was significantly associated with seropositivity on univariate analysis (Mann-Whitney U, p = 0.016, S1 Fig).

## Cross-reactivity of LDV-G reactive sera between phylogroup B ledanteviruses

LDV antisera cross-reacts with Keuraliba virus (KEUV) in complement fixation assays [12,13]. We therefore investigated whether reactivity to LDV-G by ELISA could be explained by serological cross-specificity to related viruses rather than exposure to LDV. We generated VSVΔG*luc* pseudotypes expressing the glycoproteins of three representative phylogroup B ledanteviruses; KEUV, Vaprio virus (VAPV) and Kern Canyon Virus (KCV), with pseudotypes expressing VSV glycoprotein as a negative control. First we determined IC50 values for sera from the index patient against each pseudotype. Consistent with prior infection serum titres were highest against VSVΔG*luc*-LDV-G (IC50 = 6864), followed by VSVΔG*luc*-KEUV-G (IC50 = 2981), VSVΔG*luc*-KCV-G (IC50 = 983) and VSVΔG*luc*-VAPV-G (IC50 = 141). There was minimal neutralising activity against VSVΔG*luc*-VSV-G (IC50 = 71).

Next, we randomly selected sera from individuals in the ANC-2016 cohort that were positive by LDV-ELISA and determined serum IC50 values against the pseudotype viruses. There was significant cross-neutralisation of pseudotypes bearing LDV-G and KEUV-G by ELISA positive sera (Fig 5A and 5B). Notably, IC50 titres against KEUV-G were in general higher than those for LDV-G (median LDV-G IC50 2121, median KEUV-G IC50 3166.4, median IC50 fold change for KEUV-G relative to LDV-G 0.52, IQR: 0.24–1.14). There was cross-neutralisation to a lesser extent of VAPV-G (median IC50 to VAPV-G 457.2, median fold change relative to LDV-G = 6.09, IQR: 2.26–9.27) and KCV-G (median IC50 against KCV-G 252, median fold change relative to LDV-G = 9.32, IQR = 4.73–18.1). As expected, pseudotypes bearing VSV-G were neutralised to the lowest extent (median IC50 106.4, IQR: 74.7–178.1, median fold change relative to LDV-G = 17.5, IQR: 8.21–32.5). To determine if there was an obvious spatial pattern differentiating the LDV-G ELISA positive sera based on their IC50 against LDV-G or KEUV-G, the LDV/KEUV fold change in neutralisation was expressed as log2 fold change (Fig 5C). There was no obvious pattern consistent with geographical restriction of sera with higher IC50 to either LDV or KEUV.

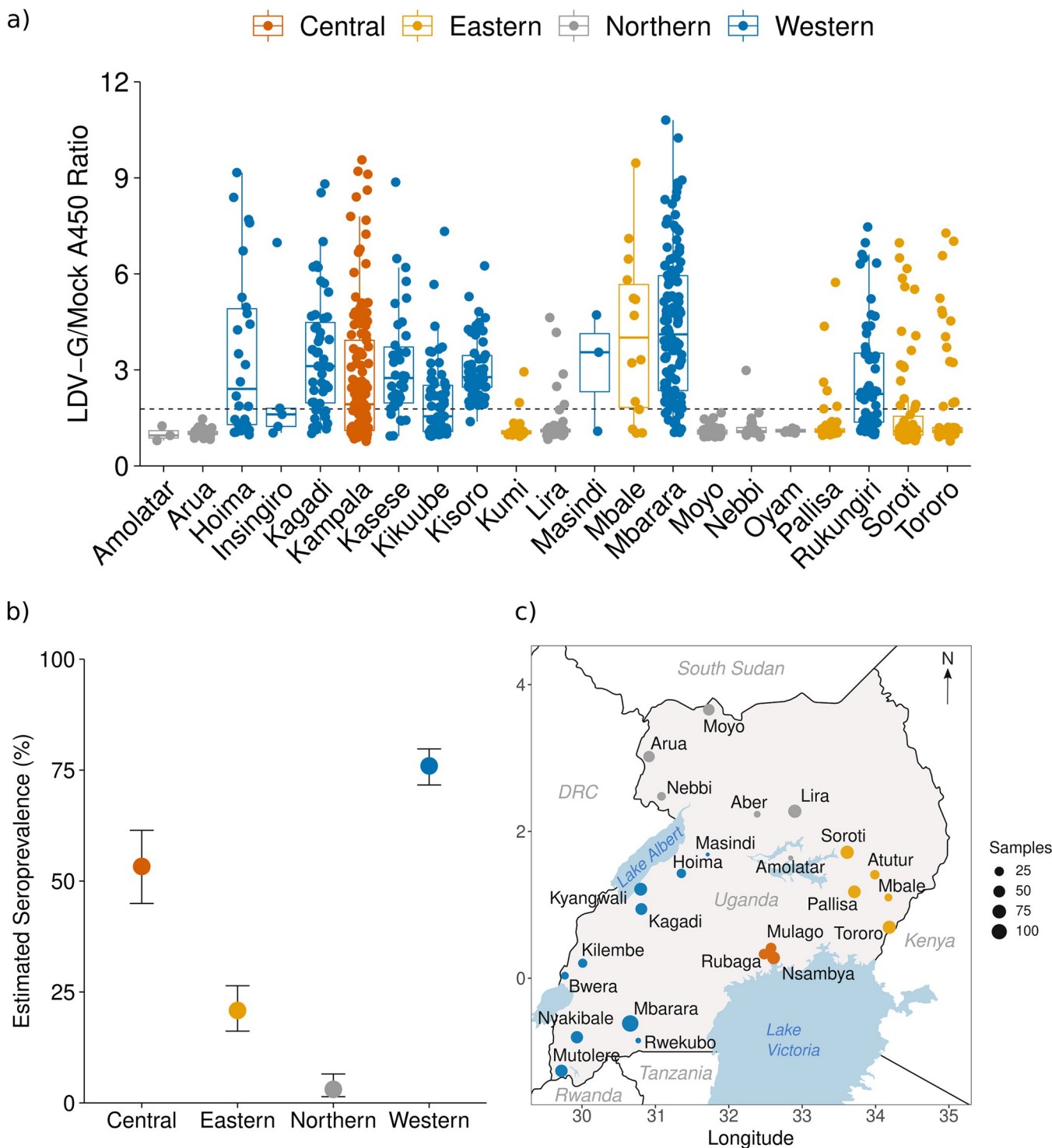

**Fig 4. Seroprevalence of ledanteviruses in Uganda by LDV-G ELISA.** (a) Raw A450 index values by district of Uganda derived from in-house LDV-G capture ELISA. Colours represent the regions of Uganda. A450 ratio values were derived by dividing the absorbance value of the test well by that of the paired mock well for each sample. Data points represent the mean value for a single individual derived from three independent experiments. Centre bars represent the median, box edges the IQR, and vertical lines the range of values within 1.5 times the interquartile range. The dashed line represents the positive cut-off derived from the mean + 3 × the standard deviation of samples derived from the West Nile subregion. (b) Estimated seroprevalence of LDV across the major regions of Uganda based on in-house LDV-G ELISA. Seroprevalence is displayed as the proportion of positive cases +/- 95% confidence intervals based on the following sample sizes: Central n = 137, Eastern n = 240, Northern n = 196, Western n = 424. (c) map of Uganda and surrounding countries and major waterbodies showing locations of study sites from the ANC-2016 study. Coloured circles represent locations of study sites with the size of each circle corresponding to the

number of participants from each site tested by the LDV-G ELISA. Maps were made in R using shapefiles from Natural Earth https://www.naturalearthdata.com/.

### Environmental correlates of LDV-G seropositivity in Uganda

As there was evidence of significant regional variation in seropositivity by LDV-G ELISA within Uganda, the influence of environmental variation at the ANC-2016 study sites was investigated by general linear mixed modelling (GLMM). Climatic (relating to rainfall and temperature), geographical (elevation and tree cover) and livestock density variables were determined for each study site. Multicollinear variables were identified and excluded and a binomial GLMM constructed with the effect of recruitment site controlled by its inclusion as a random effect. The age of individual participants was included as an explanatory variable. Model selection based on AICc showed patient age (OR 1.04 p = 0.004), forest cover (OR 1.89 p = 0.016) and the climatic variable isothermality, which describes daily and seasonal temperature fluctuations, (OR 4.81 p <0.001) were positively associated with seroprevalence (Table 1). Notably, no effect of livestock density was observed.

### Evidence of a putative novel ledantevirus in Northern Uganda

Whilst LDV has only been detected in humans, KEUV was originally isolated from various rodent species in Senegal [12]. To investigate ecological reservoirs of viral pathogens in Uganda, we performed live rodent trapping in Arua district and used mNGS to investigate the blood virome of captured rodents. Over 2233 trap nights we captured 205 rodents (S3 Table). We detected genomic evidence of a putative novel rhabdovirus in blood from a multimammate mouse, *Mastomys erythroleucus* trapped in Arua district in 2019. Host species was confirmed by sequence determination of the mitochondrial cytochrome B gene (S2 Fig). Sequencing yielded 1383706 reads of which 584 mapped to viruses of different species within the genus *Ledantevirus* by blastx. Target enrichment sequencing of the NGS library yielded a further 1296041 reads, of which 11301 mapped to ledanteviruses. *De novo* assembly of the combined reads from metagenomic and target enriched sequencing yielded 13 contigs corresponding to fragments of the nucleoprotein, phosphoprotein, glycoprotein and RDRP of a rhabdovirus with high similarity to members of the genus *Ledantevirus* (S4 Table). Further iterative *de novo* assembly using these contigs as scaffolds yielded 88% coverage of the putative nucleoprotein, 75% of the phosphoprotein, 72% of the glycoprotein, and 89% of the RDRP with reference to the closest related sequence on the NCBI database. Phylogenetic analysis placed these fragments into phylogroup B of the genus *Ledantevirus*, alongside LDV and KEUV (Fig 6). Pairwise amino acid distances in the G and L proteins suggest the *Mastomys erythroleucus*-associated ledantevirus (MELV) may possibly be assigned to a novel species within *Ledantevirus*, however confirmation of this will require completion of the full viral genome (S5 Table).

In addition to the canonical rhabdovirus genes, there was evidence of a possible U1 protein, present in other phylogroup B viruses, based on the presence of a 159 bp open reading frame terminating before the L intergenic region with no identity to rhabdovirus glycoprotein genes. Unfortunately, there was incomplete coverage of this region to permit full characterisation of this ORF or the terminal sequence of the G protein. As well as coding sequences, there was adequate sequencing coverage of gene junction sequences for the N-P, P-M and U-L intergenic regions, demonstrating the conserved rhabdovirus intergenic transcription termination-polyadenlyation (3'-ACUUUUUU-5'), and transcription initiation sequences (3'-UUGUnnUAG-5') [14]. Consistent with other phylogroup B ledanteviruses, the polyadenylation signal at the

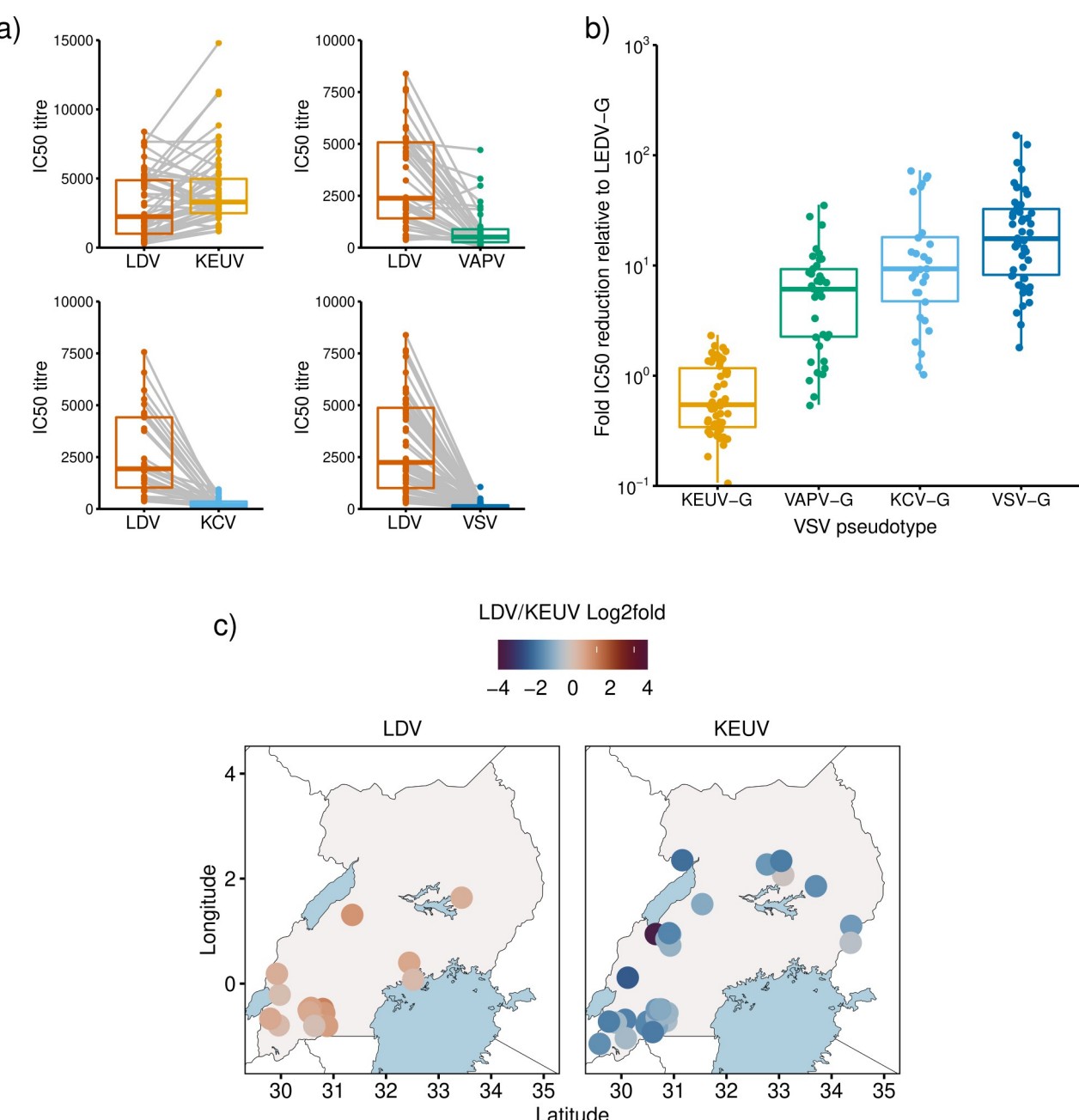

**Fig 5. LDV-G ELISA positive serum exhibits serological cross-reactivity within *Ledantevirus*.** a) IC50 titres by viral pseudotype for ANC-2016 serum samples positive by LDV-G ELISA. Between group differences were tested by paired Wilcoxon test. a; n = 60, median LDV IC50 2121, median KEUV IC50 3166.4, p < 0.001. b; n = 46, p <0.001. c; n = 37, median LDV IC50 1863, median KCV IC50 252, p <0.001. d; n = 60, p <0.001. Centre bars represent the median, box edges the IQR, and vertical lines the range of values within 1.5 times the interquartile range. Data points represent the IC50 titre of each patient sample as determined by 4-parameter logistic regression curves derived from 2 biological replicates. Grey lines link the titres for an individual against different pseudotypes. B) Fold reduction in IC50 titre of sera against each VSVΔ*Gluc* pseudotype relative to VSV(LDV-G). VSV(KEUV-G) (n = 52, median = 0.54, IQR 0.33–1.14), VSV(VAPV-G) (n = 38, median = 6.09, IQR = 2.26–9.27), VSV(KCV-G) (n = 29, median = 9.32, IQR = 4.73–18.1) and VSV(VSV-G) (n = 52, median = 17.5, IQR = 8.21–32.5). Significance was determined by Kruskal-Wallis rank-sum test (H = 111.31, df = 3, p < 0.001). c) Relative distribution of sera neutralising major *Ledantevirus* species in Uganda. Colour intensity represents log2 fold change in neutralising titre (LDV/KEUV IC50) for ELISA positive samples. Panel A; Samples with LDV titre greater than KEUV (n = 18). Panel B; Samples with KEUV titre greater than LDV (n = 34). Data points represent individual patients. Exact locations are jittered to avoid overlaps. Maps were made in R using shapefiles from Natural Earth https://www.naturalearthdata.com/.

**Table 1. GLMM of LDV-G seropositivity by patient age and ecological variables.**

|  | Estimate | Std. err | Z value | Pr(>\|z\|) | OR | 95% conf int. |
|---|---|---|---|---|---|---|
| (Intercept) | -1.59483 | 0.45667 | -3.492 | <0.001 | 0.21 | 0.08–0.50 |
| Age | 0.04232 | 0.01472 | 2.875 | 0.004 | 1.04 | 1.01–1.07 |
| Forest cover | 0.63832 | 0.26530 | 2.406 | 0.016 | 1.89 | 1.10–3.33 |
| Isothermality | 1.57128 | 0.31730 | 4.952 | <0.001 | 4.81 | 2.63–9.93 |

Marginal $R^2$: 0.466, Conditional $R^2$: 0.596

P-M junction incorporates the termination codon of the upstream phosphoprotein ORF (S3 Fig).

## Discussion

We describe the second confirmed case of human infection with LDV and demonstrate evidence of high population exposure to LDV or closely related ledanteviruses in Western Uganda. In addition, we report genomic evidence of a putative novel phylogroup B ledantevirus in blood from a *Mastomys erythroleucus*. *Ledantevirus* is an expanding genus within the family *Rhabdoviridae*, with 20 members currently designated in the latest ICTV taxonomy update [15]. The genus is divided into three phylogroups based on molecular similarity. Ledanteviruses have a close association with bats, with many species having been isolated or identified based on genomic detection either directly from bats or from their close ectoparasites.

The link to bats is strongest for the phylogroup C viruses, all of which are bat-associated. Several of these viruses have been isolated or detected in Africa. Mount Elgon bat virus (MEBV) was originally isolated from a horseshoe bat *Rhinolophus hildebrandtii* in Kenya near the border with Uganda [16,17]. Experimental studies demonstrated that MEBV is capable of infecting mosquitoes, but onwards transmission to laboratory animals was not demonstrated. The closely related Kanyawara virus (KYAV) was isolated from haematophagous insect parasites of bats (bat flies) sampled from *Myonycteris* species in Western Uganda [18]. Subsequently, KYAV was detected in bat flies sampled from a colony of *Lissonycteris angolensis ruwenzorii* bats near the Uganda-DRC border [19]. There was heavy infection of bat flies by KYAV, with 20 of 26 insects yielding genomes of KYAV by mNGS and a further insect harbouring the genome of another closely related virus, Bughendera virus (BUGV). There was significant genomic diversity between the ledanteviruses detected in this study, despite being contemporaneously sampled from insects in the same bat colony. Phylogenetic analysis of KYAV genomes combined with mitochondrial DNA sequences derived from their insect hosts suggested KYAV may be primarily maintained in bat flies by vertical transmission. Notably there was also evidence of infection of host bats, with one individual exhibiting detectable virus in saliva, suggesting a possible role of bats as amplifying hosts facilitating occasional horizontal transmission of KYAV between bat flies.

Kumasi rhabdovirus (KRV) was isolated from the spleen of *Eidolon helvum* sampled from a large bat colony in Ghana, with subsequent RNA detection by PCR in organs of 5.1% of 487 bats surveyed [20]. Viral RNA was rarely detected in the lungs or intestines in these bats, whereas spleen tissue was positive in all samples, with the inference that infection was confined to the circulatory and lymphoid system and that respiratory or gastrointestinal viral shedding would be unlikely. Antibodies against KRV were detectable by IFA in 11.5% of bats and 11% of humans residing in the area near the bat roosts, indicating either exposure to a common vector species or direct transmission from bats to other mammals potentially through

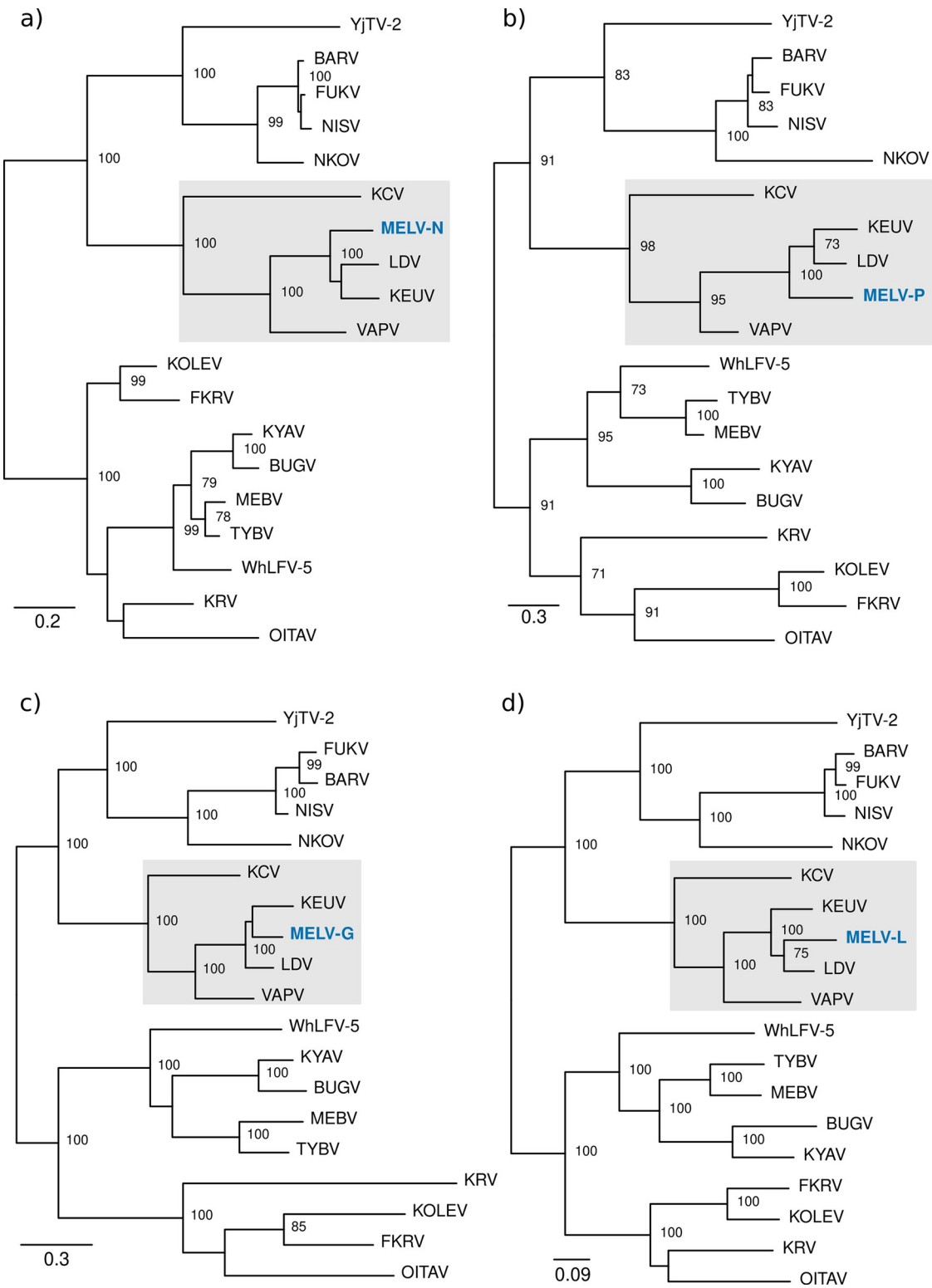

**Fig 6. Phylogenetic placement of the *Mastomys erythroleucus* associated ledantevirus genomic fragments within *Ledantevirus*.**
(a) a continuous 373 amino acid fragment corresponding to the MELV nucleoprotein. (b) three continuous fragments of 87aa, 72aa and 73aa corresponding to the MELV phosphoprotein. (c) a continuous 411 amino acid fragment corresponding to the MELV glycoprotein. (d) five fragments of 65aa, 120aa, 529aa, 814aa and 168aa corresponding to the MELV L protein. For each gene alignments were created based on complete coding sequences of all members of the genus *Ledantevirus* and gene fragments of

MELV and maximum likelihood phylogenies generated. Node labels represent bootstrap support values for nodes with support >70 based on 1000 replicates. The grey highlighted areas represent the phylogroup B viruses. MELV; *Mastomys erythroleucus*-associated ledantevirus. LDV; Le Dantec virus, KEUV; Keuraliba virus, VAPV; Vaprio virus, KCV; Kern Canyon virus, YjTV-2; Yongjia tick virus 2, BARV; Barur virus, FUKV; Fukuoka virus, NISV; Nishimuro virus, NKOV; Nkolbisson virus, WhLFV-5; Wuhan louse fly virus 5, TYBV; Taiyi bat virus, MEBV; Mount Elgon bat virus, BUGV; Bughendera virus, KYAV; Kanyawara virus, FKRV; Fikirini rhabdovirus, KOLEV; Kolente virus, KRV; Kumasi rhabdovirus, OITAV; Oita virus.

consumption of bat meat or exposure to dead or injured bats. Notably cattle, sheep and goats from the same area displayed no serological evidence of exposure, however 27% of swine, which would be expected to feed on injured or dead bats encountered whilst foraging, were positive. Several other viruses form a clade with KRV within phylogroup C. Fikirini rhabdovirus was isolated from the liver of a *Hipposideros vittatus* captured in Southern Kenya [21]. In this case the virus was cultured from lung, kidney, brain, and intestine as well as faecal swabs. Kolente virus was isolated in Guinea from both ticks and a *Hipposideros* bat [22]. Beyond Africa, Oita virus was isolated from blood of a horseshoe bat *Rhinolophus cornutus* in Japan in 1972. Tayai bat virus was detected by mNGS in brain tissue of *Rhinolophus sinicus* sampled in China in 2007 [23]. Both Wuhan louse fly virus 5, derived from an unidentified hippoboscid fly sampled from a bat in China, and Wenzhou Rhinolophus pusillus ledantevirus 1, derived from unspecified samples from horseshoe bats in China, have been detected using mNGS, but detailed descriptions of the circumstances of their detection have not been published [24,25].

Phylogroup A ledanteviruses appear less closely associated with bats that the bat-associated phylogroup C. In several cases individual viruses in this group have been identified in both biting arthropods and vertebrates, indicating they are likely to be vector-borne. Fukuoka virus (FUKV) was initially isolated in Japan from biting midges *Culicoides punctatus* and *Culex tritaeniorhynchus* mosquitoes with evidence of neutralising antibody in cattle housed in the area of arthropod collections [26]. Several years later FUKV was again isolated in Japan, this time from cattle serum drawn from febrile sentinel surveillance animals [27]. Barur virus (BARV) has been isolated from both rodents and ticks in India, and on several occasions from *Rhipicephalus* ticks in Africa where it has recently been detected by mNGS of ticks parasitising Ugandan cattle [28,29]. A virus serologically indistinguishable from BARV has also been isolated from *Rhipicephalus pulchellus* ticks in Kenya, however molecular identification was not performed in this case [30]. Nkolbisson virus was originally isolated in 1964 from mosquitoes in Cameroon and was subsequently isolated from human samples in the Central African Republic, although there are few details available of its association with human disease [31,32]. Nishimuro virus was isolated from serum from a healthy wild boar in Japan. More recently, metagenomic studies have identified further members of phylogroup A in mainland China in rodents (Longquan Niviventer coninga ledantevirus 1) [24] and ticks, with both Tongren rhabd tick virus and Yongjia Tick Virus 2 detected by mNGS of pooled *Haemaphysalis* ticks [25].

The phylogroup B ledanteviruses have all been isolated from tissue or serum of vertebrates. Vaprio virus was isolated from several organs derived from a single *Pipistrellus kuhlii* bat collected near a bat roost in Italy [33]. Live virus was cultured from homogenised lung and heart tissue, and other organs including brain and kidneys were positive by PCR, suggesting active infection of this animal. Bat ectoparasites were not collected in this study, however PCR testing of other bats revealed presence of the VAPV RNA in pooled organs from two further bats; *P. kuhlii* and *Hypsoyo savii*. KEUV was isolated from several rodent genera in Senegal. The initial isolation of Kern Canyon virus (KCV) was from pooled spleen and heart derived from a *Myotis yumanensis* bat captured in California [34]. LDV was originally isolated from a 10-year-old girl who presented to hospital in Senegal in 1965 with fever [9,35,36]. A further possible case was identified based on complement fixation assays in a Welsh dock worker who was bitten by

an unspecified arthropod in 1969 whilst unloading a ship that had travelled from West Africa. Several days later he developed fever, headache and encephalopathy, requiring admission to hospital for six weeks [37]. Virus was not isolated in this case, but serological testing performed several years later in 1976 showed antibody reacting with LDV by complement fixation. Given the cross-reactivity demonstrated between KEUV and LDV in this and other studies, the evidence for definite human infection with LDV in this case in the absence of virus isolation is relatively weak.

Metagenomic sequencing has been increasingly applied in cases of unidentified acute febrile illness in Africa, leading to detection of several novel viruses including rhabdoviruses [8,38]. Determining the clinical significance of viral species detected by sequencing alone can however be challenging [39]. Demonstration of seroconversion or detection of viral antigens in diseased tissue by histology can support the role of a novel agent as a pathogen [40,41], but the clinical samples required for such analyses are often unavailable and can be challenging to access in low-resource settings. Tissue sampling for histological examination is particularly difficult to obtain in non-fatal cases. Thus, it is often unclear if sporadically detected agents such as LDV represent pathogens or are incidental to an alternative disease process. The recent detection of two novel orthobunyaviruses in Uganda, both clearly associated with febrile illness but for which serological or other pathological parameters have not yet been reported, are illustrative of this challenge [6,7]. Even when serological investigations have been performed, clear determination of pathogenicity can be difficult.

In illustration of these difficulties several novel rhabdoviruses of the genus *Tibrovirus* have been detected in human clinical samples in Africa [38,42]. Of these, Bas-Congo virus (BASV) is notable for its association with a cluster of 3 cases of acute haemorrhagic fever [8]. The BASV genome was discovered by metagenomic sequencing of acute plasma from the one surviving case in this cluster. Whilst there was strong serological evidence of exposure to BASV in this individual, a subsequent prospective sero-epidemiolocal study demonstrated evidence of high population exposure to BASV in the area it was initially isolated, challenging its role as a high-consequence pathogen and a cause of haemorrhagic fever [43]. The other tibroviruses detected in humans have not been associated with similarly severe presentations. The Ekpoma viruses (EKV-1 and EKV-2) were detected by mNGS from the blood of healthy individuals in Nigeria. Subsequent population serology has indicated exposure to both viruses may be common in West and Central Africa [38,43]. Similarly, the only described case of Munduri virus (MUNV), a tibrovirus isolated from a patient with nodding syndrome in South Sudan was associated with a community seroprevalence of 22% without a significantly elevated seroprevalence in cases of nodding syndrome, indicating MUNV was unlikely to be cause of the patients presentation [42]. Such cases highlight the need for careful design of studies employing mNGS for viral discovery with the inclusion of appropriate control populations. Systematic screening for novel or emerging pathogens by mNGS at sentinel clinical sites or through national surveillance systems combined with appropriately designed seroprevalence studies may allow accurate estimation of their contribution to local infectious disease epidemiology and inform the need for development of diagnostic platforms for use beyond research settings.

LDV has been exclusively detected in association with febrile illness and can be classified as a pathogen in humans. To support this, we have shown a specific serological response by the patient directed against LDV-G, although owing to unavailability of samples from the acute phase of illness we were unable to demonstrate a change in neutralising titre suggestive of seroconversion. The patient tested negative against a panel of pathogens commonly causing febrile illness in this setting. Additionally, we report widespread population exposure to either LDV itself or other members of the genus *Ledantevirus* in Uganda, particularly in the Western Region where the patient was recruited. The high seroprevalence in Western Uganda suggests

ledanteviruses may predominantly cause subclinical or mild infection and that more severe cases such as that described here go unreported or undiagnosed. Notably the patient in this case made a full recovery. Such viruses are often able to spread undetected by surveillance systems in low and middle-income countries as demonstrated by the emergence and expansion of Zika virus [44]. As we detected LDV in a sample originally collected in 2012, and our serological data is based on samples from 2016, further prospective studies are required to determine the current contribution of ledanteviruses to AFI in Uganda.

Most of the LDV-G ELISA positive sera in the ANC-2016 cohort tested against a panel of phylogroup B ledantevirus glycoproteins had higher IC50 titres against KEUV-G than LDV-G, suggesting that in addition to LDV, human infection by closely related ledanteviruses may be occurring in Uganda. Serological cross-reactivity between LDV and KEUV was first noted in the 1980s [12], with the close serological relationship mirrored by relatively high similarity at the amino acid level [35]. Consistent with the serological data suggesting further members of the LDV serogroup may be causing human infection in Uganda, we detected the partial sequence of a close relative of KEUV and LDV by mNGS of blood sampled from a *Mastomys erythroleucus* in Arua district. The glycoprotein of this virus displayed 76.7% amino acid identity with KEUV and 75.6% with the LDV glycoprotein. This putative novel ledantevirus is therefore likely to exhibit cross-reactive behaviour in serological assays with both LDV and KEUV. Unfortunately, our study is limited by our inability to determine the complete glycoprotein sequence of this virus, meaning we were unable to test for evidence of exposure in the ANC-2016 cohort by pseudotype-based neutralisation. Genomic species demarcation for the genus *Ledantevirus* include a protein amino acid sequence divergence of 7% in the L gene and 15% in the glycoprotein. The MELV L protein diverged from the LDV L protein by 18.4% based on 89% genomic coverage of the putative L protein, and 24.1% based on 72% coverage of the putative glycoprotein. Whilst the coding-complete genome is required for confirmation, based on the genomic information available this virus may represent a new species within phylogroup B of the genus *Ledantevirus*.

Ledantevirus seroprevalence as measured by LDV-G ELISA was geographically heterogenous within Uganda, suggesting either regional variation in the presence or prevalence of infection within reservoir or vector species or differences in the dynamics of human exposure. Notably, the MELV, a potential driver of human LDV seroprevalence in Uganda, was discovered in Arua district where human LDV-G seroprevalence was low, raising doubt as to its contribution to the seroprevalence estimates based on the LDV-G ELISA. A reservoir species for LDV has not yet been identified, but seroprevalence was highest in the Western Region. This area is notable for the isolation of the bat-associated phylogroup C ledanteviruses KYAV and BUGV [18,19]. The Western Region, including Kasese district where the LDV case was identified, has been identified as having high ecological suitability for the presence of bat species and is a high risk area for spillover of bat-associated viruses into human populations [45]. However, it is unlikely that either KYAV or BUGV are drivers of the seropositivity by LDV-G ELISA in this area. Our pseudotype-based neutralisation assay suggested that cross-reactivity among LDV-G ELISA positive serum was limited to a subset of phylogroup B ledanteviruses, with minimal evidence of cross-neutralisation of more distantly related viruses within phylogroup B including KCV. Additionally, previous studies have failed to demonstrate serological cross-reactivity between MEBV, a phylogroup C rhabdovirus closely related to KYAV and BUGV and either LDV, KEUR or KCV [13,17,22]. It has been demonstrated that bat flies can host diverse ledanteviruses within a single bat colony [19]. Given the diverse bat species present in Uganda, and the spatial distribution of seropositivity in the human population, it remains plausible that LDV or related phylogroup B viruses are associated with yet unsampled bats or their ectoparasites in this area.

The MELV is the second virus in phylogroup B of ledantevirus to be detected in rodents. KEUV was initially isolated from various rodent genera in Senegal in 1968 including *Mastomys* and gerbils of the genera *Taterea* and *Taterillus* [36]. There is strong evidence that ledanteviruses are vector borne, with several viral species being isolated from both vertebrates and biting arthropods, as well as evidence of maintenance of viral lineages through transovarial transmission [18,22,28,31]. Whilst the detection of viral RNA in the serum of this mouse may represent ingestion of an infected invertebrate, there have been multiple documented examples of ledantevirus infection of various mammal species with virus isolation from blood and other tissues, making active infection of the rodent in this case plausible [33,34]. It remains unclear as to whether the rodent was incidentally infected or is part of a more established host-vector-virus relationship. Metagenomic NGS of mosquito and livestock–associated tick pools collected in Arua concurrent to the rodent collections described in this paper revealed no evidence of ledanteviruses [29,46], suggesting that further studies should focus on close ectoparasites of rodents such as ticks and fleas.

Our analysis of ecological conditions suggests forest cover and isothermality are associated with increased LDV-G seropositivity in Uganda. Notably, the Western Region of Uganda is home to several forest parks with a wide diversity of vertebrate species, including bats and rodents [47,48]. Stable climatic conditions may permit greater wildlife densities throughout the year and could act to increase population exposure to zoonotic disease in these areas. Communities based in forested areas have also been demonstrated consume relatively more bushmeat. Alternatively, isothermality and forest cover may simply be correlated with the presence of a greater diversity of wildlife and the presence of a reservoir species restricted to these areas.

There are limitations in both our population based seroprevalence study and the environmental variables we employed for analysis of environmental correlates of LDV-G seropositivity. We used stored convenience samples from an antenatal study conducted in 2016 for the LDV-ELISA based seroprevalence study, limiting our sample to women of childbearing age and meaning that several key risk groups were not covered. For example, the practice of hunting of wild animals for food carries a risk of exposure to zoonotic pathogens. Rodents and bats are often hunted by male children and adults, which were not present in our sample [49–51]. In addition, there was limited individual-level metadata associated with our sample, with no information on potentially significant variables such as occupation or exposure history. Thus, extrapolation of this study to the wider population may lead to inaccurate estimates of seroprevalence. In parallel to the serosurvey, potentially important variables in the environmental analysis including the density of potential reservoir species such as bats were not readily available and were not included. Factors including the presence and density of potential reservoir hosts and human behavioural factors including bushmeat hunting are likely to be important in mediating the extent of human infection with zoonotic viruses. Future prospective studies should be designed to be representative of the population structure in Uganda and associated with detailed individual-level metadata in combination with a specific set of environmental variables.

We used a VSV pseudotype-based assay neutralisation assay to detect neutralising antibody directed against ledanteviruses in human serum. This system may be less specific than live virus neutralisation, potentially leading to inflated seroprevalence estimates or exaggerating the presence of cross-neutralisation. However, pseudotype-based neutralisation assays have been shown to correlate well with authentic virus neutralisation assays and have been used extensively for the study of rhabdovirus glycoproteins [52–54]. Additionally, the serum derived from the LDV index patient demonstrated an appropriate pattern of neutralising antibody, with neutralisation of pseudotypes bearing LDV-G to high titre without a significant effect on those bearing VSV-G, indicating limited non-specific neutralisation.

In summary we identified Le Dantec virus (species *Ledantevirus ledantec*) as the likely causative agent in a case of acute febrile illness in Western Uganda and demonstrated serological evidence of widespread exposure to phylogroup B ledanteviruses in the human population of the Western, Central and Eastern regions of the country. A phylogroup B ledantevirus was detected in blood from a common and geographically widespread rodent species *Mastomys erythroleucus*. Identification and characterisation of the infectious agents which routinely spillover into human populations, providing opportunities for emergence and evolution within human hosts is important for epidemic preparedness. Further characterisation of the epidemiology of acute febrile illness and prospectively designed population level serosurveys are required to establish the precise contribution of ledanteviruses to human disease in Uganda. The strong association with bats across the genus *Ledantevirus* and our detection of a phylogroup B ledantevirus in a *Mastomys erythroleucus* indicates further studies investigating the reservoir species of ledanteviruses in Uganda should focus on bats, rodents, and close ectoparasites of both.

## Methods

### Ethics statement

Ethical approval for the AFI study, the use of stored serum samples from the ANC-2016 cohort and for wild rodent collections in Arua district was granted by the Research Ethics Committee of the Uganda Virus Research Institute with reference numbers GC/127/10/02/19 (human sampling, February 2010), GC/127/18/09/662 (human and animal sampling, September 2018) and GC/127/19/06/662 (human and animal sampling, June 2019. Research approvals were granted by the Uganda National Council for Science and Technology (UNCST, reference numbers HS2485 and HS767).

### Clinical samples

Written informed consent was obtained from all patients. Acute serum from the patient was collected as part of the acute febrile illness study [10]. A further episode of sampling for additional sera from this individual occurred in 2018. Samples from the ANC-2016 HIV surveillance study were collected in 2016 from individuals who had provided informed consent for the future use of their samples for medical research. ANC-2016 samples were supplied by Robert Downing (UVRI).

### Rodent sampling

Rodent sampling and euthanasia was conducted in line with international best practice guidelines [55]. Rodent trapping was performed in three villages in Adumi subcounty; Ombaci, Oniba, and Sua, in April 2019. Rodents were trapped in Sherman traps (H.B. Sherman Trap Company) and Tomahawk traps (Tomahawk Live Trap Company) baited with peanut butter mixed with smoked fish and a piece of sweet potato. Traps were set in four locations in relation to homesteads: indoors (inside sleeping or cooking huts), the peri-domestic compound area (within five metres of the sleeping and cooking huts), the peri-domestic bush (within five metres of the homestead edge), and sylvatic (up to 300 metres into the bush from the centre of the village). Two Sherman and two Tomahawk traps were placed within the peri-domestic compound area. An additional three Sherman and three Tomahawk traps were placed in the peri-domestic bush setting. To sample the sylvatic environment one Tomahawk and one Sherman trap were placed every 20 metres for 300 metres from the centre of the village in each of the four cardinal directions (North, East, South, West).

Traps were set before dusk and opened shortly after dawn. Trapping was conducted over three nights in Ombaci and Oniba and for two nights in Sua for a total of 2233 trap nights. Seven traps were lost during the sampling period. Rodents were euthanised in the field by inhalation of halothane and cervical dislocation, followed immediately by blood sampling by heart puncture. Animals were speciated using taxonomic keys. Blood samples were centrifuged and sera heat inactivated and stored at -80˚C until further use.

### Next generation sequencing

For human and rodent serum samples RNA was extracted from 200 µl of plasma using the Agencourt RNAdvance Blood Kit (Beckman Coulter) according to the manufacturer's instructions and eluted in nuclease-free water. RNA was reverse transcribed using Superscript III (Invitrogen) followed by dsDNA synthesis with NEBNext (New England Biolabs). DNA libraries were prepared using LTP low-input Library preparation kit (KAPA Biosystems). Resulting libraries were quantified with the Qubit 3.0 fluorometer (Invitrogen) and their size determined using a 4200 TapeStation (Agilent). Libraries were pooled in an equimolar ratio and sequenced on the Illumina MiSeq platform using 150x2 v2 cartridges.

Target enrichment of RNA derived from rodent blood was performed using custom-designed biotinylated RNA probes targeting full genomes of all known arboviruses, including ledanteviruses (Agilent) using manufacturer's instructions. Briefly, dual-indexed DNA library was incubated overnight at 65˚C. Captured DNA was recovered using Streptavidin T1 Dynabeads, washed with provided buffers, and eluted in water. Captured library was amplified using Illumina primers and sequenced on an Illumina NextSeq 500.

### Bioinformatic analysis

Adaptor trimming and read quality control was performed with trim_galore [56] using a minimum read length of 80 bp following adaptor removal and a minimum Phred score of 20. Sequence reads were classified using diamond version 0.8.20.82 in blastx mode [57] and visualised using Krona [58]. *De novo* assembly of the human serum sample was performed with SPAdes version 3.11.1 [59]. The *de novo* contigs were screened using diamond blastx and confirmed by blastn against the NCBI nr database. The *de novo* assembly was then used as a reference to align raw sequencing reads using the short read aligner tanoti (https://github.com/vbsreenu/Tanoti). For sequencing of rodent serum *denovo* assembly was performed using the metavic pipeline [60] and contigs mapping to *Ledantevirus* species were determined by diamond blastx. The genome organisation of ledantevirus contigs was manually determined by approximating their position based on their alignment by blastx against either the LDV or KEUV reference genomes. The resulting construct was used as a scaffold for further iterative *de novo* assembly with Gapfiller version 1–11 [61]. Cytochrome B sequence barcoding for speciation of *Mastomys* was performed by aligning NGS sequencing reads to the optimal *Mastomys* mitochondrial reference sequences, with the optimal reference determined by blastn query of initial consensus sequence against the NCBI nucleotide database (MZ131552). The subsequent consensus was aligned with a panel of previously published reference sequences [62].

### Phylogenetic analysis

To determine the phylogenetic placement of the LDV-Uganda L protein sequence recovered from Ugandan patient, full-length L protein amino acid sequences from representative rhabdovirus taxa within the *Alpharhabdovirinae* were aligned. For phylogenetic placement of partial amino acid sequences of the putative N, P, G and L genes of the novel rhabdovirus derived

from AV_R184 the corresponding full-length amino acid sequences for each gene from all members of the *Ledantevirus* were aligned. All alignments were created with mafft version 6.240 using the L-INS-i method [63]. Poorly aligned regions were removed using trimAl version 1.2 [64]. Phylogenetic trees were inferred in RAxML version 8.2.10 [65] using the LG substitution model [66] with an invariable site plus discrete Gamma model of rate heterogeneity across sites as determined by Modelfinder implemented in IQ-TREE version 1.6.12. Substitution models were selected based on the AIC criterion [67]. Support for each node was determined using 1000 bootstrap replicates. Trees were visualised using figtree version 1.4.4 (http://tree.bio.ed.ac.uk/software/figtree/).

## Nested RT-PCR amplification

Plasma RNA was extracted from 140 µl plasma aliquots using QIAamp Viral RNA Mini Kit (Qiagen Cat No 52904). Next, cDNA was generated in a 30 µl reaction containing RNA, reverse primer LEDPOL-51R (5'- CAACGCACATATCCTTCATCATCAGC-3') and master mix reagents (SuperScript IV (SSIV) Reverse Transcriptase Kit, ThermoFisher 18090050) following manufacturer's recommendations. Nested PCR amplification was carried out using Platinum Taq DNA Polymerase High Fidelity (ThermoFisher cat No 11304029) and the following primer sets aimed to amplify a 5'-half genome sequence of the metagenomic-inferred LDV strain. This fragment contains nucleoprotein, phosphoprotein, matrix, glycoprotein genes, and a small polymerase subgenomic fragment.First round amplification performed with forward primer LEGA5-F (5'-TTTTCTGGTCTTCTCTTTTTCCTACTGAAA-3') and reverse primer LEDPOL-51R (see sequence above) generates a 5,706 bp fragment. Second round PCR amplification carried out with forward primer LDAN-F (5'- ATGGCTAACGAGA CAATTTATCGTTTCTC-3') and reverse primer LEDPOL-52R (5'- GATGCATTCAACAT CAACACCATGTCATG-3') generates a 5,464 bp fragment. A PCR positive control included an RNA extract from the Le Dantec viral isolate DakHD763 obtained from an infected Senegalese child in 1965. DakHD773, a BSL-2 rhabdovirus was imported as a lyophilized cultured supernatant derived from an experimentally infected mouse, as a kind donation from Dr. Thomas Ksiazek from the University of Texas Medical Branch, Galveston, TX 77555). Experimental positive PCR reactions visualized in agarose gel electrophoresis were sequenced by Sanger chemistry using ABI Big dye Terminator v3.1 kit and ABI 3500 Genetic Analyser (ThermoFisher) per manufacturer's recommendations.

## Le Dantec real-time RT-PCR

RNA was extracted from 200µl of plasma using the Agencourt RNAdvance Blood Kit (Beckman Coulter) or the Qiagen viral minikit (see above) according to the manufacturer's instructions. Extracted RNA was tested directly for Le Dantec virus by real-time PCR or stored at -80˚C until use. An in-house developed real-time RT-PCR assay targeting a 145 bp fragment of the LDV glycoprotein gene was develop as a highly sensitive detection assay of the Le Dantec virus in clinical samples. An aliquot of RNA was mixed in a 30 µl reaction volume containing reagents from SuperScript III Platinum One-Step Quantitative RT-PCR System (ThermoFisher Cat No. 11732–020) 0.2 mM forward primer LEDAG-F (5'- GCTTGAAATGCCCTG AAGCT-3'), 0.2 mM reverse primer LEDAG-R (5'-TCACATCTRGTCAACCATCTTGA-3') and 0.1 mM TaqMan probes labelled with 6- carboxyfluorescein (FAM) LEDAG-FAM (5'-CCATCACCCTATGTTCATCAGGACC-3') in the presence of 0.05 µM ROX Reference Dye. RT-PCR conditions were those recommended by the manufacturer for 50 cycles of amplification in an ABI 7500 Real-Time PCR System (Applied Biosystems).

## LDV-G ELISA

The development of the LDV-G ELISA has been described previously [10]. Briefly, the LDV-G transmembrane domain was identified using TMHMM [68]. The predicted ectodomain of LDV-G was cloned into the secretory mammalian expression vector pHLSec containing a C-terminal 6xHistidine tag. Human embryonic kidney cells (HEK 293T), maintained in Dulbecco modified Eagle's medium supplemented with 100 IU/ml penicillin, 100 μg/ml streptomycin, 2 mM glutamine and 10% foetal bovine serum were transfected using Fugene 6 (Promega) and cell supernatant containing secreted LDV glycoprotein was harvested at 48 hours post transfection. Presence of His-tagged protein was confirmed by Western Blotting against 6xHis.

ELISA plates (Thermo Scientific) were coated with rabbit polyclonal anti-6xHis antibody (Abcam) diluted to 1:1000 in basic coating buffer and incubated overnight at 4˚C. The next day plates were blocked with 2.5% BSA in PBS at 37˚C for 1 hour. Cell supernatant containing LDV-G-His or mock protein was applied to the wells followed by human serum diluted 1:50 in blocking buffer or purified IgG for 1 hour at 37˚C. This was followed by goat anti-human IgG peroxidase antibody (Sigma) at room temperature for 1 hour. Wells were washed after each step with 0.1% Tween-20 in PBS. The ELISA reaction was developed using TMB substrate (Thermo Scientific), stopped with 0.16M sulphuric acid and absorbance read at 450 nm using a Pherastar FS plate reader (BMG Labtech). Each sample was tested in a test well containing the truncated LDV-G-His protein as the capture antigen and, to identify nonspecific binding, a mock well with supernatant derived from mock-infected cells transfected with empty plasmid vector. The test result was reported as the OD450 reading from the sample test well divided by that of the sample mock well to give a test/mock OD450 ratio for each sample. For each serum sample the ELISA was repeated on at least three occasions on separate plates. Each plate included known positive and negative control samples, as well as a blocking buffer only sample. All patient samples were heat-inactivated by incubation at 60˚C for 30 minutes before use in the assay.

## Production of VSVΔ*Gluc* pseudotypes

For expression of the KEUV, KCV and VAPV glycoproteins (KEUV-G, KCV-G and VAPV-G), complete coding sequences were derived from GenBank accession numbers NC_034540.1, NC_034451.1, and NC_043538.1 and synthesised with the addition of a SalI restriction site and Kozak sequence immediately before the 5' initiation codon and a NotI restriction site immediately following the 3' stop codon (GeneWiz/BioBasic). For expression of LDV-G the glycoprotein sequence derived from the Ugandan patient was codon optimised and synthesised (Eurofins genomics). Coding sequences of LDV-G, KEUV-G, VAPV-G and KCV-G were then cloned into the multiple cloning site of mammalian expression vector VR1012 using the SalI and NotI restriction sites, or the NotI and BglII restriction sites in the case of LDV-G, to create VR1012-LDV-G, VR1012-KEUV-G, VR1012-VAPV-G and VR1012-KCV-G. Plasmid pMDG was used to express VSV-G.

For each pseudotype virus $2 \times 10^6$ HEK293T cells were seeded into 10 cm dishes in 10 ml DMEM supplemented with 10% FCS. The next day cells were transfected with the relevant glycoprotein expression plasmid using Fugene 6 (Promega) as per the manufacturer's instructions. One 10cm dish was not transfected to serve as a no glycoprotein control. Plates were incubated overnight at 37˚C in a 5% $CO_2$ atmosphere. The next day plates were removed from the incubator and infected with 15ul of $1.15 \times 10^7$ TCID50/ml VSVΔ*Gluc*-VSV-G (corresponding to an approximate MOI of 0.03) and returned to the incubator for 3 hours. Media was discarded and cells washed three times with warmed PBS before the addition of 10 mls DMEM

with 10% FCS. The cells were re-incubated for a further 72 hours after which supernatants containing pseudotype virus were harvested, passed through a 0.45 μm filter (Starlab) and stored at -80°C before further use.

## Pseudotype-based neutralisation assay

HEK293T cells were grown in a 75 cm$^2$ flask until 70% confluent. Media was removed and cells washed with 2 ml trypsin before being resuspended in 8 ml DMEM + 10% FBS. Cells were then plated at a density of $5\times10^4$ cells per well in a volume of 50 μl per well in each well of a white 96-well plate.

Serum samples were serially diluted in DMEM supplemented with 10% FBS. Pseudotype preparations were mixed 1:1 with each dilution and incubated at 37°C for 30 minutes. 50 μl of the pseudotype/sera mixture was added to the plated cells in triplicate for a final dilution series ranging from 1:32 to 1:131,072 and a "no sera" control well. Plates were incubated at 37°C, 5% CO$_2$ for 24 hours before addition of 75 μl Steadylite luciferase substrate solution (Perkin Elmer) to each well. Plates were incubated at room temperature for 10 minutes before luminescence was read using a Chameleon V luminometer (Hidex). Luciferase activity for each dilution was derived from the mean of three plate replicates normalised to the average for the no sera control wells. Percentage neutralisation for each dilution was calculated by subtracting the normalised luciferase activity from 1 and multiplying by 100. The IC50 value for each serum sample and pseudotype combination was determined by interpolation of a four-parameter logistic regression curve between percentage neutralisation and the reciprocal serum dilution. Regression curves were fitted in R version 3.6.3/R Studio version 1.2.5033 using the package drc version 3.0–1 with an upper constraint of 100 using a minimum of 2 biological replicates to fit each curve.

## Immunocytochemistry

The LDV-G sequence was cloned into the expression vector VR1012 with the addition of a C-terminal 6xHistidine tag to generate VR1012-LDV-G-His. Coverslips (Fisher Scientific) were added to each well of a 24-well plate (Corning) and 0.5 ml 70% IMS added to each well for 20 minutes. IMS was removed and coverslips allowed to air dry before being washed 3 times with 1ml PBS. BHK-21 cells were plated at a density of $1\times10^5$ per well and incubated at 37°C, 5% CO$_2$ until 70% confluent. Cells were transfected with VR1012-LDV-G-His using Fugene 6 transfection reagent (Promega) and returned to the incubator for 24–48 hours.

Media was removed and cells fixed with 4% formaldehyde before permeabilization with 200 μl 0.1% Triton-X-100 per well for 5 minutes. Cells were then washed three times with 1000 μl PBS. The primary antibody mixture comprised a rabbit polyclonal to 6X His tag antibody (Abcam) at a dilution of 1:500 mixed with human serum at a dilution of 1:200 with BSA 1% as the diluent (ThermoFisher Scientific). The primary mixture was added to the cells at a volume of 200 μl per well and incubated overnight at 4°C. The next morning cells were washed three times with PBS and blocked with 1% BSA for 30 minutes. The secondary antibody mixture was a combination of an Alexa Fluor 488-conjugated goat anti-rabbit IgG secondary antibody (Invitrogen) at a concentration of 4 μg/ml and an Alexa Fluor 594-conjugated goat anti-human IgG secondary antibody (Invitrogen) at a concentration of 5 μg/ml diluted in 1% BSA. The secondary antibody mixture was added to wells at a volume of 200 μl and the plate placed on a shaker at room temperature for 45 minutes before three further 5 minute washes with 1000 μl PBS. Coverslips were stained with Hoechst 33342, at a concentration of 1:2000 in PBS for 15 minutes. Coverslips were then washed three times with 1000 μl PBS and once with ultra-pure water before being mounted on glass slides with 2 μl CitiFluor AF1 Mountant solution

(Electron Microscopy Sciences). Images were acquired using a LSM 710 confocal microscope (Carl Zeiss Microscopy), and images processed using ZEN Blue software (Carl Zeiss Microscopy).

### Environmental analysis

The R package raster version 3.4–5 was used to extract bioclimatic, geographical and live-stock data at the locations of the ANC-2016 study sites. Bioclimatic variables were derived from the Worldclim dataset [69]. Tree cover data was extracted from the global forest change dataset [70]. Livestock density data was extracted from the UN FAO dataset (https://livestock.geo-wiki.org/home-2/). Elevation was extracted from the STRM 90m DEM Digital Elevation Dataset (https://srtm.csi.cgiar.org/). Variables were averaged across a 10km radius around each study site. Multicollinearity between variables was tested by creating a correlation matrix by the method of Spearman and subsequently a hierarchal clustering dendogram to identify highly correlated groups of variables. A correlation coefficient cut-off of 0.6 was used to group variables. One variable from each group was selected to be included in the global model.

Binomial generalised mixed models were constructed in R version 3.6.3/R Studio version 1.2.5033 using package lme4 version 1.1.26. In addition to bioclimatic variables, patient level age was included in the analysis. To identify the optimal random effects structure an initial model was created combining all fixed effects terms and a random effects term of either site, region or a nested random effect (region/site), with the optimal random effect term selected based on the AICc criterion [71]. The best fitting model was then identified using the dredge function package MuMIn version 1.43.17 [72], with the AICc criterion used to select the final model. The response variable was patient level positivity on LDV ELISA, with the final fixed effect explanatory variables being patient age, forest cover, and isothermality (BIO3), with a random effect of site (24 levels). Continuous response variables were normalised. The model was called as glmer(seropositivity) ~ age + scale(forest_cover,center = (mean forest cover))) + scale(isothermality,center = (mean isothermality)) + (1|site), family = binomial). Model assumptions were tested using the R package DHARMa version 0.4.4 [73]. Conditional and marginal R2 measures were computed using R package MuMIn.

### Inclusion statement

The research was conducted as part of a partnership between the MRC-University of Glasgow Centre for Virus Research, the MRC/UVRI & LSHTM Uganda Research Unit and the Uganda Virus Research Institute. The study was designed and conducted by researchers based in Uganda and the UK. Field and laboratory work were performed by trained individuals following a full health and safety risk assessment. Field PPE was employed when handling small mammals (rubber boots, coveralls, goggles and gloves). Laboratory samples were handled in a class 2 biological safety cabinet. The study was reviewed by the UVRI research ethics committee and the UNCST. All sampling and fieldwork protocols relating to wildlife sampling were based on established protocols in use at UVRI. All individuals meeting the criteria for authorship have been included as authors.

### Supporting information

**S1 Table. Assays employed in the acute febrile illness study (AFI).**
(DOCX)

**S2 Table. Genomic distance between LDV-Uganda and LDV-Senegal (DakHD763/ KM205006).**
(DOCX)

**S3 Table. Number of rodents by species captured for serum mNGS in Adumi subcounty.**
(DOCX)

**S4 Table. Blastx results of de novo contigs from sequencing of blood from *Mastomys erythroleucus*.**
(DOCX)

**S5 Table. Pairwise amino acid distance between the *Mastomys erythroleucus*-associated ledantevirus and other phylogroup B ledanteviruses.**
(DOCX)

**S6 Table. Source data for Fig 2.**
(XLSX)

**S7 Table. Source data for Fig 4.**
(XLSX)

**S8 Table. Source data for Fig 5.**
(CSV)

**S1 File. Source data for Fig 3 - AFI patient.**
(CZI)

**S2 File. Source data for Fig 3 –positive control.**
(CZI)

**S3 File. Source data for Fig 3 –negative control.**
(CZI)

**S1 Fig. Age by LDV-G ELISA reactivity in the ANC-2016 cohort.** Age by seroprevalence as determined by LDV-G ELISA for individuals in the ANC-2016 cohort (n = 997). Median age in the seropositive group was 25 years (n = 451, IQR = 21–30) compared to 24 in the seronegative group (n = 546, IQR = 21–29), Mann-Whitney U, p = 0.016. Centre bars represent the median, box edges the IQR, vertical lines the range (1.5 times the IQR from the upper and lower quartile), and points the outliers.
(TIF)

**S2 Fig. Species identification of *Mastomys erythroleucus* by mitochondrial cytochrome B sequencing.** Maximum likelihood phylogeny of mitochondrial cytochrome B sequences derived from representative *Mastomys* species. The sequence derived from the animal in which the *Mastomys erythroleucus* associated ledanteviruses was detected is indicated in orange. *Serengetimys pernanusis* is included as an outgroup. Circles indicate nodes with bootstrap support >70.
(TIF)

**S3 Fig. The *Mastomys erythroleucus*—associated ledantevirus virus shares conserved gene junction regions with other group B ledanteviruses.** Genomic gene junction regions of MELV compared with other group B ledanteviruses demonstrating the presence of conserved rhabdovirus intergenic polyadenylation and transcription initiation signals. a) nucleoprotein–phosphoprotein junction. b); phosphoprotein matrix protein junction. c) accessory protein–RNA dependant RNA polymerase junction. MELV; *Mastomys erythroleucus*—associated

ledantevirus, LDV; Le Dantec virus, KEUV, Keuraliba virus, VAPV; Vaprio virus, KCV; Kern Canyon virus.
(TIF)

## Author Contributions

**Conceptualization:** James G. Shepherd, Jesus F. Salazar-Gonzalez, Maria G. Salazar, Robert G. Downing, Hanna Jerome, Joseph T. Mpanga, Chris Davis, Linda A. Atiku, Josephine Bwogi, Julius J. Lutwama, Daniel G. Streicker, Pontiano Kaleebu, Emma C. Thomson.

**Data curation:** James G. Shepherd, Shirin Ashraf, Jesus F. Salazar-Gonzalez, Maria G. Salazar, Robert G. Downing, Henry Bukenya, Hanna Jerome.

**Formal analysis:** James G. Shepherd, Shirin Ashraf, Jesus F. Salazar-Gonzalez, Maria G. Salazar, Hanna Jerome, Daniel G. Streicker, Emma C. Thomson.

**Funding acquisition:** Daniel G. Streicker, Emma C. Thomson.

**Investigation:** James G. Shepherd, Shirin Ashraf, Jesus F. Salazar-Gonzalez, Maria G. Salazar, Henry Bukenya, Hanna Jerome, Joseph T. Mpanga, Chris Davis, Lily Tong, Vattipally B. Sreenu, Linda A. Atiku, Nicola Logan, Ezekiel Kajik, Yafesi Mukobi, Cyrus Mungujakisa, Michael V. Olowo, Emmanuel Tibo, Fred Wunna, Hollie Jackson Ireland, Andrew E. Blunsum, Iyanuoluwani Owolabi, Ana da Silva Filipe, Josephine Bwogi.

**Methodology:** James G. Shepherd, Shirin Ashraf, Maria G. Salazar, Vattipally B. Sreenu, Ana da Silva Filipe, Daniel G. Streicker.

**Project administration:** James G. Shepherd, Shirin Ashraf, Jesus F. Salazar-Gonzalez, Maria G. Salazar, Robert G. Downing, Henry Bukenya, Joseph T. Mpanga, Linda A. Atiku, Andrew E. Blunsum, Emma C. Thomson.

**Resources:** Robert G. Downing, Linda A. Atiku, Nicola Logan, Ana da Silva Filipe, Josephine Bwogi, Brian J. Willett, Julius J. Lutwama, Pontiano Kaleebu, Emma C. Thomson.

**Software:** Vattipally B. Sreenu.

**Supervision:** Chris Davis, Josephine Bwogi, Brian J. Willett, Julius J. Lutwama, Daniel G. Streicker, Pontiano Kaleebu, Emma C. Thomson.

**Visualization:** James G. Shepherd.

**Writing – original draft:** James G. Shepherd, Emma C. Thomson.

**Writing – review & editing:** James G. Shepherd, Shirin Ashraf, Jesus F. Salazar-Gonzalez, Maria G. Salazar, Robert G. Downing, Henry Bukenya, Hanna Jerome, Joseph T. Mpanga, Chris Davis, Lily Tong, Vattipally B. Sreenu, Linda A. Atiku, Ana da Silva Filipe, Josephine Bwogi, Brian J. Willett, Julius J. Lutwama, Daniel G. Streicker, Pontiano Kaleebu, Emma C. Thomson.

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
