## [Decision Letter · Decision Letter 0]

20 Feb 2024

Dear Dr Shepherd,

Thank you very much for submitting your manuscript "Widespread human exposure to ledanteviruses in Uganda: a population study." for consideration at PLOS Neglected Tropical Diseases. As with all papers reviewed by the journal, your manuscript was reviewed by members of the editorial board and by several independent reviewers. In light of the reviews (below this email), we would like to invite the resubmission of a significantly-revised version. 

The manuscript is essentially praised for its contribution by Reviewers 1 to 3, who are all expert reviewers. Reviewer 1 called the manuscript "intriguing" and Reviewer 3 described the manuscript as "much needed". However, Review 1 has requested very significant changes to the text in addition to the changes requested by Reviewers 2 and 3 which are summarised by the AE below.

We cannot make any decision about publication until we have seen the revised manuscript and your response to the reviewers' comments. Your revised manuscript is also likely to be sent to reviewers for further evaluation.

Sincerely,

Michael W Gaunt, PhD

Academic Editor

Michael Holbrook

Section Editor

Comments from the Academic Editor

The manuscript is essentially praised for its contribution by Reviewers 1 to 3, who are all expert reviewers. I do personally thank each reviewer for their contribution and attention to the manuscript.

Reviewer 3 makes two important points: 1) that the Ethics is incorrectly reported and must be clarified; 2) inquiring whether PRNT assays were performed as a comparison. 

All Reviewers identified significant areas of improvement of the text.

Both Review 1 and 2 have complained that the manuscript omits significant prior published investigations. Reviewer 2 has very kindly provided a reading list detailing significant published history around this virus starting in 1977 (see attachment) and must be included in the ms. Reviewer 1 was concerned as to how the findings fitted with the "scientific, historical, ecological or epidemiological context" of the virus and is a serious criticism. Thus the reading list by Reviewer 2 is starting point, albeit an extremely helpful starting point towards complying with the request of Reviewer 1. All reviewers concurred that there were no technical problems with the ms, however compliance with ICTV is required.

Please note this manuscript review turn around time was delayed by a potential reviewer agreeing to review, but not submitting a report. This always results in delay. I am personally grateful to Reviewer 3 for stepping in to ensure a timely turn around. The turn around time does not in any way adversely reflect on the quality of the manuscript.

Reviewer's Responses to Questions

**Key Review Criteria Required for Acceptance?**

**Methods**

-Are the objectives of the study clearly articulated with a clear testable hypothesis stated?

-Is the study design appropriate to address the stated objectives?

-Is the population clearly described and appropriate for the hypothesis being tested?

-Is the sample size sufficient to ensure adequate power to address the hypothesis being tested?

-Were correct statistical analysis used to support conclusions?

-Are there concerns about ethical or regulatory requirements being met?

Reviewer #1: See below

Reviewer #2: Objectives, study design, sampling and statical analyses are fine.

No ethical concerns.

Reviewer #3: include the ethics approval dates for all the human and animal sampling protocols included in the manuscript

**Results**

-Does the analysis presented match the analysis plan?

-Are the results clearly and completely presented?

-Are the figures (Tables, Images) of sufficient quality for clarity?

Reviewer #1: See below

Reviewer #2: Results and their presentation, including figures and tables are fine.

Reviewer #3: yes

**Conclusions**

-Are the conclusions supported by the data presented?

-Are the limitations of analysis clearly described?

-Do the authors discuss how these data can be helpful to advance our understanding of the topic under study?

-Is public health relevance addressed?

Reviewer #1: See below

Reviewer #2: Conclusions are fine.

Reviewer #3: yes

**Editorial and Data Presentation Modifications?**

Reviewer #1: See below

Reviewer #2: Some modifications to data presentational required.

Reviewer #3: (No Response)

**Summary and General Comments**

Reviewer #1: This intriguing manuscript describes a new case of ledantec virus (LDV) infection in humans (at least the third documented case, not the second, as the authors claim) and a serosurvey of antibodies to LDV and its relatives in Uganda. The serologic assays are impressive and the authors should be commended for their efforts to investigate human infection.

Unfortunately, the manuscript fails to place its findings in an appropriate scientific, historical, ecological or epidemiological context. There is a significant literature on these viruses, including in Uganda, which the authors do not cite or merely cite cursorily. As a result, many of the main conclusions drawn by the authors are deeply in error, based on known facts from the published literature. 

The authors should “start from scratch.” They should step back and familiarize themselves with the fascinating history and biology of the ledanteviruses. They should read and understand all the research on these viruses that has preceded their work (it’s a rich literature but not intractably extensive). They should then rewrite the manuscript entirely, structuring it around the current state of knowledge. An incomplete list of essential features of this restructuring includes:

1) Where, in nature, other ledanteviruses have been found.

2) A detailed review of the strong evidence showing that ledanteviruses are vector-borne, including which vectors have been associated with which viruses. This is true even for ledanteviruses for which no vector has ever been identified.

3) A review of how ledanteviruses are thought to be transmitted among mammalian hosts by vectors, maintained in mammalian hosts by vectors, and occasionally transmitted to humans by those vectors, including documented examples going back many decades.

4) Which ledanteviruses have previously been discovered in Uganda, where, when, and under what circumstances. The authors do not cite a number of key papers from several research groups about other ledanteviruses in Uganda, including zoonotic transmission.

5) Which ledanteviruses have previously been discovered in other African countries. Note Kenya, for example, where Mt Elgon Bat virus was found – Mt. Elgon lies on the border between Uganda and Kenya.

6) Which ledanteviruses have previously been discovered in countries outside of Africa (ledanteviruses are found around the world), and how our knowledge of these viruses in aggregate paints a fairly clear picture of the biology, ecology and epidemiology of these viruses globally.

The authors are strongly advised to seek the expert opinion of a specialist in ledanteviruses as they rewrite their manuscript. This would help the authors form more accurate conclusions about their own data. 

Line 22 and 179. This appears to be a cross-sectional sample. Please delete “a cohort of.”

Line 34. Change “pandemics” to “epidemics” to avoid alarmist language.

Line 34-36. The grammar of this sentence is confusing. What is the hotspot? Significant gaps? Africa? Please re-write sentence to be clearer.

Line 46. “Environmental reservoir” is not accurate. Either delete “environmental” or change to “animal reservoir.”

Lines 49-66. The opening paragraphs are sweeping and overly general. Many of the statements are also false. For example, the epidemiology of acute febrile illness in sub-Saharan Africa is comparatively well characterized – even though non-malarial etiologies are not. There’s really no meaningful difference between diagnosis of febrile illness in developing African countries versus more developed countries – virtually all acute fevers in both settings are treated empirically and without a laboratory diagnosis, with the exception of malaria and COVID.

Figure 1. The phylogenetic tree includes 16 members of the genus Ledantevirus. However, there are currently 21 recognized ledanteviruses. All should be included, especially given the emphasis of this paper on the viral genus and not just LDV. The full list of ledantevirus member species can be found in the ICTV ledantevirus chapter: https://ictv.global/report/chapter/rhabdoviridae/rhabdoviridae/ledantevirus. The authors will note that coding complete genomes are available for all 20 currently recognized member species and for Tongren rhabd tick virus 2, which is currently unclassified but which is also a ledantevirus and should be included too. This chapter also includes very useful summary information about the natural history of these viruses (see “Biology” subheading). This would be a good (although not complete) starting point for the literature review mentioned above.

Figure 1. Similar to the above comment, there are 32 recognized alpharhabdovirine genera, but only 20 are included in the tree (see here: https://ictv.global/report/chapter/rhabdoviridae/rhabdoviridae ). However, the other alpharhabdovirine genera do not add to the story presented here, and in fact the addition of outgroups reduces phylogenetic resolution. The tree would be more informative with a single outgroup – I would suggest vesicular stomatitis Indiana virus as the exemplar species of the genus Vesiculovirus. Thus, a new tree should consist of all current members of the Ledantevirus genus and VSIV as the outgroup.

Line 179. Biases associated with including only women of childbearing age in the serosurvey should be discussed fully in the Discussion. This is a major limitation affecting the validity of the study, and it does not adhere to the human subjects policies of the NIH regarding sex and age representation.

Lines 263-274 and 682-687. It is unclear why these particular environmental variables were assessed, based on current knowledge about the ecology and epidemiology of the ledanteviruses. The authors should re-do this analysis using variables that are carefully chosen to test hypotheses about the reservoirs and transmission of ledanteviruses. For example, such variables could include: bat density (or proxy measures such as the proximity of caves); small mammal biodiversity; the density of reported acute fevers of unknown origin, etc. These sorts of ecological analyses can be informative, but only if the variables are carefully selected to test highly specific hypotheses about modes of transmission. Otherwise, as is the case currently, the analyses tend to uncover broad factors that are difficult to interpret because they are non-specific and confounded.

Line 278. “Environmental” is confusing because it sounds as if the virus was floating in a lake or found on a log. Change to “A ledantevirus in a wild rodent.” See next comment.

Lines 279-303. The finding of fragmentary ledantevirus genetic material in fewer than 1% of captured rodents is very similar to what has been found for other ledanteviruses and other rhabdoviruses. The likely explanation is that the virus was not, in fact, infecting the mouse, but rather that it was infecting an insect with which the mouse interacted. This might have been an ectoparasite of the mouse (e.g. a flea, louse, mite, etc). Carriage of ledanteviruses in ectoparasites is a central feature of the natural history of the ledanteviruses. The presence of fragmentary viral RNA in the blood of the mouse would thus indicate transient or low-level infection. It is equally likely that the mouse merely ingested an ectoparasite (“allogrooming”) or ate a non-parasitic invertebrate infected by the virus. In the latter case, viral RNA appeared in the blood of the mouse because oral ingestion of viruses leads to viral genetic material in blood in small mammals (this is documented for many viruses, including rhabdoviruses and ledanteviruses). Concluding that the rodent is infected from the data available would repeat a common error made in the study of these and other viruses. This is an example of why a careful and comprehensive review of the literature would be necessary for proper interpretation of data in this study.

Figure 6. The new virus does not meet the ICTV species demarcation criteria for the genus Ledantevirus: https://ictv.global/report/chapter/rhabdoviridae/rhabdoviridae/ledantevirus. In other words, it may be a variant of LDV or KEUV and not a novel virus at all. The authors should delete any text referring to this as a novel virus throughout the manuscript. They should give the virus a placeholder name such as “Ledantevirus from Mastomys erythroleucus” but not use the name “Odro virus” or any other such label. This is important because of the great difficulty of un-doing the erroneous assignment of novel viral taxon names, which is unfortunately pervasive and leads to endless confusion in viral taxonomy. In the Discussion, the authors should mention the species demarcation criteria for the Ledanteviruses and state that more complete genome sequencing would be required to ascertain whether this virus is a putative new member species of the genus or a variant of a currently recognized member species. If, in the future, the authors do indeed generate such sequence data, they should submit a Taxonomic Proposal to ICTV, who will then evaluate whether the virus merits assignment as a novel species.

Line 340-342. It is fascinating that the authors found LDV in an East African patient, and the serological data are intriguing. These data could be very useful and interesting for the field.

Lines 342-344. Delete, as the data d

---

## [Decision Letter · Decision Letter 1]

22 May 2024

Dear Dr Shepherd,

Thank you very much for submitting your manuscript "Widespread human exposure to ledanteviruses in Uganda: a population study." for consideration at PLOS Neglected Tropical Diseases. As with all papers reviewed by the journal, your manuscript was reviewed by members of the editorial board and by several independent reviewers. The reviewers appreciated the attention to an important topic. Based on the reviews, we are likely to accept this manuscript for publication, providing that you modify the manuscript according to the review recommendations. 

The manuscript has been solidly reviewed and we wish to thank the reviewers for their time and expertise. All reviewers are clear that the authors' have addressed their original critiques. The authors' must revise the text in accordance with the requests made by Reviewer 1 and 2 prior publication, these are mostly minor edits. Please note, Reviewer 2's report is a downloadable document.

Sincerely,

Michael W Gaunt, PhD

Academic Editor

Michael Holbrook

Section Editor

Reviewer's Responses to Questions

**Key Review Criteria Required for Acceptance?**

**Methods**

-Are the objectives of the study clearly articulated with a clear testable hypothesis stated?

-Is the study design appropriate to address the stated objectives?

-Is the population clearly described and appropriate for the hypothesis being tested?

-Is the sample size sufficient to ensure adequate power to address the hypothesis being tested?

-Were correct statistical analysis used to support conclusions?

-Are there concerns about ethical or regulatory requirements being met?

Reviewer #1: See below

Reviewer #2: No further comments

Reviewer #3: I have reviewed this manuscript prior to its transfer to PLoS NTD and the authors have successfully addressed all my comments. Nothing more to add

**Results**

-Does the analysis presented match the analysis plan?

-Are the results clearly and completely presented?

-Are the figures (Tables, Images) of sufficient quality for clarity?

Reviewer #1: See below

Reviewer #2: No further comments

Reviewer #3: (No Response)

**Conclusions**

-Are the conclusions supported by the data presented?

-Are the limitations of analysis clearly described?

-Do the authors discuss how these data can be helpful to advance our understanding of the topic under study?

-Is public health relevance addressed?

Reviewer #1: See below

Reviewer #2: line 1023 - a species (abstraction) cannot cause illness. Only the virus (animate) can cause illness. Should read: "We identified Le Dantec virus (species Ledantevirus ledantec [italics]) as the likely causative agent...."

Reviewer #3: (No Response)

**Editorial and Data Presentation Modifications?**

Reviewer #1: See below

Reviewer #2: No further comments

Reviewer #3: (No Response)

**Summary and General Comments**

Reviewer #1: Many thanks to the authors for their careful consideration of all the reviewer comments and for re-writing sections of the text. The review of ledanteviruses in Uganda and elsewhere is good. I have no major suggested further changes. Please consider the following minor changes.

Line 34. Add “(Rhabodoviridae)” in italics after “Ledantevirus.” This will make the author summary stand alone for those who may not realize that Ledantevirus is a rhabdoviral genus.

Line 58. Check with journal as to whether “Next Generation Sequencing” should be capitalized. None of the words are proper nouns.

Line 237. It’s not entirely clear what “spatial restriction” means. Consider an alternative term.

Lines 271-302. This is a long paragraph and it’s difficult to digest all the information as a block. Consider splitting it into two shorter paragraphs, for example with a paragraph break at line 287 and some minor corresponding rearrangements of text.

Line 374. The Hippoboscidae are actually not bat flies. They are “louse flies” or “keds.” Bat flies are in different families. Change to “hippoboscid fly.”

Line 406. Change “female” to “woman” because, following the previous sentence, it sounds as if a female bat presented to hospital.

Lines 415-449. This is again a very long paragraph. There is a natural paragraph break at line 428, if a transition statement such as “In illustration of these difficulties” were added before “Several novel rhabdoviruses.”

Line 451. The word “probable” may be too cautious here. The evidence is quite strong of LDV being a pathogen in humans (but not bats or arthropods). So, remove “probable” and change line 451 to “a pathogen in humans.”

Line 518. Consider changing “framework” to “relationship.”

Reviewer #2: No further comments

Reviewer #3: (No Response)

PLOS authors have the option to publish the peer review history of their article (what does this mean?). If published, this will include your full peer review and any attached files.

Reviewer #1: No

Reviewer #2: No

Reviewer #3: No

Figure Files:

Data Requirements:

Reproducibility:

References

---

## [Editor Report · Decision Letter 2]

17 Jun 2024

Dear Dr Shepherd,

We are pleased to inform you that your manuscript 'Widespread human exposure to ledanteviruses in Uganda: a population study.' has been provisionally accepted for publication in PLOS Neglected Tropical Diseases.

Before your manuscript can be formally accepted you will need to complete some formatting changes, in addition to those listed by the AE below, which you will receive in a follow up email. A member of our team will be in touch with a set of requests.

Best regards,

Michael W Gaunt, PhD

Academic Editor

Michael Holbrook

Section Editor

This is robust work, however all references have formatting errors with around one third of references having multiple errors. The authors might have output the contents of Mendeley or EndNote directly without editing the bibliography. The errors are summarised in the AE notes. A copy of the list has been forwarded to journal staff. Also note the authors' omitted the report detailing their correction described by in the attachment by Reviewer 2, therefore the authors' are asked to review the document to confirm all changes were implemented.

# AE notes

1. Mixing title capitalisation of each word in the title (e.g. reference 5 and 20), with first word capitalisation.

2. Failure to italicise any paper titles in the ms when referring to a virus, e.g. reference 61 for de novo and 49 on mouse Lineman taxonomy, also reference 62

3. Mixing full journal names, with internationally recognised abbreviations. The is a 50:50 split. For example "Journal of Virology. " reference 20 with "J Gen Virol." reference 22

4. I don't the "The American Journal of Tropical Medicine ..." is a journal.

5. Missing page number ranges in many references, e.g. reference 2, 11, 18, 19, 23 ... 45

6. References with page number ranges omit are truncated for the final page for example reference 21 "2393-8."

7. Inconsistent naming of journals, for example reference 5 reads "PLOS ONE", elsewhere the authors write "PLoS One", other times the authors write "PLOS Tropical Neglected Diseases" (reference 38) and PLoS Curr. (reference 45)

8. Missing journal volume numbers, for example reference 43

9. Several references have a "[" character within the title, which is not closed, i.e. no "]", but should not be there to start with, for example reference 15, 32, 56

10. Reference 47, appears to be a book without a publisher, year?, relevant chapter or page numbers and authors of the chapter

11. To expand on point 3 above reference 52 the journal is "Front Immunol." but in reference 53 it is "Frontiers in Microbiology" (with no page numbers).

12. Reference 61 "Suppl 14" is repeated twice sequentially

13. Journal title " Bioinformatics (Oxford, England). ", the publishers location is removed for a journal.

14. Reference 72 is incorrect, this an R package and should be referenced as reference 73.

15. Please check the reference format of the paper requested by Reviewer 2 their point 3 below.

Note the point Reviewer 1 raised about "Next Generation Sequencing" capitalisation mid-sentence, the authors responded that they would ask journal staff. The authors have a title regarding "Next generation sequencing" with first word capitalisation only and is therefore inconsistent with the mid-sentence statement.

Finally, the corrections report by the authors omitted the attachment of corrections provided by reviewer 2. All corrections checked have been resolved, however again I'm requesting the authors check all points which have been copy and pasted below.

1. The author’s fail to acknowledge a previous report of putative Le Dantec virus infection in a dock worker from Wales who had not been out of Britain for 20 years but had been bitten by an insect in 1969 whilst unloading a ship from Nigeria. (Woodruff et al BMJ 1977). Although evidence of this infection was based only on CF antibody against Le Dantec virus, it should be noted in the paper.

2. The original description of the isolation of Le Dantec virus was in 1968 in a report from Paul Brès of the Reference Centre for Arboviruses at the Pasteur Institute in Dakar. This report is in French and somewhat difficult to access but it is referenced in the International Catalogue of Arboviruses (Berge, 1975) which is now curated online by the CDC. It would be appropriate to make some reference to this original report in addition to the secondary report of Cropp et al (1985) which has been cited by the authors.

3. Ledanteviruses of phylogroup B differ from members of other phylogroups in that they include a small ORF (U1) in an additional transcriptional unit between the G and L genes. It would be useful if the authors indicated whether this ORF was detected in the genome sequence of Odro virus.

4. The authors refer (line 366) to two novel orthobunyaviruses from Uganda that have been associated with febrile illness. One wonders why the authors do not refer to multiple tibroviruses (family Rhabdoviridae) that have also been associated with febrile illness in central Africa. Bas Congo virus from the Democratic Republic of Congo (Grard et al PLoS Path 2012), Mundri virus from South Sudan (Edridge et al Viruses 2016), Ekpoma virus 1 from Nigeria and Ekpoma virus 2 from Nigeria and Angola (Stremlau et al PLoS NTD 2015; Kuhn et al Viruses 2020) have each been detected by NGS in humans, in some cases in association with febrile illness. The difficulties in establishing causal relationship for Bas Congo virus and other such viruses identified by metagenomic sequencing has also been discussed previously by these and other authors. Reference to human infection in central Africa with these other rhabdoviruses would be a useful addition to the discussion.

5. Taxonomic nomenclature is expressed incorrectly throughout the manuscript. Guidelines for correct taxonomic terminology and usage are described in detail on the ICTV website (https://ictv.global/filebrowser/download/440) and elsewhere (Zerbini et al Arch Virol 167:1232, 2022; Walker et al Animals 12:1363, 2022).

Line 18 The authors state “Le Dantec virus (LDV), type species of the genus Ledantevirus…”. Firstly, ICTV no longer recognises type species but instead user exemplar members of a species. Secondly, viruses are not species but are assigned as members to a species. Le Dantec virus is assigned taxonomically as a member of the species Ledantevirus ledantec, genus Ledantevirus, family Rhabdoviridae. A paper describing the difference between viruses (concrete entities) and virus species (abstract entities of human construction) can be found on the ICTV website.

This would be correctly expressed as “Ledantec virus (LDV), assigned to the species Ledantevirus ledantec, genus Ledantevirus, family Rhabdoviridae…..”

Line 25 Should be “…was confined to ledanteviruses,…” (no caps , no italics).

Line 28 Should be “Ledantevirus infection…” (no italics – caps only as the start of a sentence).

Line 77 Should be “…detection of a novel ledantevirus,..” (no caps , no italics)

Line 105 Correctly expressed (caps and italics) as this refers to the taxon, not the virus(es).

Line 287 Should be “..mapped to viruses of different species within the genus Ledantevirus by blastx.”

Line 288 Should be “..mapped to ledanteviruses.” (no caps, no italics)

Line 291 Should be “..mapped to members of the genus Ledantevirus.” (caps and italics)

Line 295 Should be “…of the genus Ledantevirus,..”

Line 296-7 Should be “…suggest Odro virus can be assigned to a novel species within the genus Ledantevirus”

Line 301 Correctly expressed (no caps and no italics) as this refers to the viruses, not the taxon.

Line 317 Should be “…members of the genus Ledantevirus,..”

Line 345 Should be “…members of the genus are divided into three phylogroups” (phylogroups are not taxa but are phylogenetic clusters of viruses).

Line 351 Should be “…members of the genus Ledantevirus,..”

Line 359 Should be “…viruses of different species detected by sequencing..” (viruses [concrete entities] can be sequenced but species [abstract entities] cannot be sequenced)

Line 379/80 Should be “…members of the genus Ledantevirus,..”

Line 401 Should be “genera Taterea and Taterillus.”

Line 406,444 Should be “…ledantevirus..” (no caps, no italics)

Line 503 Should be “…ledanteviruses…” (no caps, no italics)

Line 515,516 Should be “…ledantevirus..” (no caps, no italics)

Line 529 Should be “..the Alpharhabdovirinae..”

Line 532 Should be “…members of the genus Ledantevirus,..”

---

## [Editor Report · Acceptance letter]

1 Jul 2024

Dear Dr Shepherd,

We are delighted to inform you that your manuscript, "Widespread human exposure to ledanteviruses in Uganda: a population study.," has been formally accepted for publication in PLOS Neglected Tropical Diseases.

Best regards,

Shaden Kamhawi

co-Editor-in-Chief

Paul Brindley

co-Editor-in-Chief
